# High-throughput single-cell rheology in complex samples by dynamic real-time deformability cytometry

Bob Fregin [1], Fabian Czerwinski [1], Doreen Biedenweg[2], Salvatore Girardo [3], Stefan Gross [2,4], Konstanze Aurich [2] & Oliver Otto [1,4]

In life sciences, the material properties of suspended cells have attained significance close to that of fluorescent markers but with the advantage of label-free and unbiased sample characterization. Until recently, cell rheological measurements were either limited by acquisition throughput, excessive post processing, or low-throughput real-time analysis. Real-time deformability cytometry expanded the application of mechanical cell assays to fast on-the-fly phenotyping of large sample sizes, but has been restricted to single material parameters as the Young's modulus. Here, we introduce dynamic real-time deformability cytometry for comprehensive cell rheological measurements at up to 100 cells per second. Utilizing Fourier decomposition, our microfluidic method is able to disentangle cell response to complex hydrodynamic stress distributions and to determine viscoelastic parameters independent of cell shape. We demonstrate the application of our technology for peripheral blood cells in whole blood samples including the discrimination of B- and CD4+ T-lymphocytes by cell rheological properties.

[1] Zentrum für Innovationskompetenz: Humorale Immunreaktionen bei kardiovaskulären Erkrankungen, Universität Greifswald, Fleischmannstr. 42, 17489 Greifswald, Germany. [2] Universitätsmedizin Greifswald, Fleischmannstr. 8, 17489 Greifswald, Germany. [3] Biotechnology Center, Center for Molecular and Cellular Bioengineering, Technische Universität Dresden, Tatzberg 47/49, 01307 Dresden, Germany. [4] Deutsches Zentrum für Herz-Kreislauf-Forschung e. V., Standort Greifswald, Universitätsmedizin Greifswald, Fleischmannstr. 42, 17489 Greifswald, Germany. Correspondence and requests for materials should be addressed to O.O. (email: oliver.otto@uni-greifswald.de)

With the potential for label-free phenotyping of cellular states and functions, the mechanical properties of cells have gained an increasing importance over the last years[1–3]. Being sensitive to cytoskeletal and nuclear alterations, this biomarker has been used to track the stability, passaging, and differentiation of stem cells, to follow the activation of immune cells, and to characterize metabolic states[4–8]. As mechanical phenotyping is based on intrinsic cell material properties, it serves as a complementary approach to traditional molecular biology methods and is of an increasing importance in fundamental and applied research, where molecular markers are not wanted or not available. However, a broad translation of mechanical phenotyping into life science applications had so far been hampered by lack of a fast and robust measurement technique. While traditional methods like atomic force microscopy, micropipette aspiration, and optical stretching were limited to analysis rates of less than 100 cells per hour[9–11], the introduction of microfluidic concepts increased the throughput by several orders of magnitude[12,13]. The serial deformation of cells in a hydrodynamic environment allows for throughput rates on the order of 100–10,000 cells per second, which is a prerequisite for screening applications, e.g., the combination of biophysical and molecular analysis or the characterization of highly potent skeletal stem cells in regenerative medicine[14,15].

In contrast to well established cell biology techniques, like flow cytometry, the parameter space of mechanical cell characterization cannot simply be extended by additional molecular markers, but is limited to any information that can be extracted from acoustical, mechanical, or optical measurements[16–18]. However, cells are far away from a thermal equilibrium. Their response to an external mechanical load in the form of creep or stress relaxation is highly nonlinear and driven by both, an active and a passive intrinsic remodeling, which has to be explored to link cytoskeletal properties to cell function[19–21]. While rheological experiments and the determination of a frequency-dependent complex modulus have initially been performed on adherent cells[2,22], microfluidic systems in combination with high-speed video microscopy enabled an increase in throughput and an extension to suspended cells[23,24].

Using a parallel array of micron-sized constrictions, Lange et al. utilize the confinement of suspended cells in a microfluidic channel to estimate cell elasticity and fluidity from flow speed, residence time, and driving pressure. Power-law rheology explains the collapsing of data from multiple cell lines and under multiple conditions onto a master curve and is in agreement with the theory of soft glassy materials[25,26]. Quantitative deformability cytometry extends this concept by introducing calibrated microspheres to extract quantitative information and allows for potential comparison to reference methods like micropipette aspiration[27].

In contrast to micro-constrictions, methods like deformability cytometry (DC), real-time deformability cytometry (RT-DC) and real-time fluorescence and deformability cytometry (RT-FDC) are contactless and utilize solely hydrodynamic stress to deform cells[24,28,29]. In addition, RT-DC and RT-FDC are capable to perform image acquisition and analysis on-the-fly, which allows for a label-free screening of heterogeneous cell samples of virtually unlimited size and the identification of sub-populations based on mechanical properties. However, in real-time data analysis, image acquisition and data evaluation have been limited to a single snapshot per cell and, thus only steady-state material parameters as the Young's modulus can be derived[30,31].

Here, we introduce dynamic RT-DC (dRT-DC) for single cell rheological measurements in heterogeneous samples where we capture the full dynamics of suspended cells passing the central constriction of a microfluidic channel on-the-fly. We show that Fourier analysis of cellular shape modes allows to disentangle the complex cell response to time-dependent and time-independent hydrodynamic stress distributions, which are typical for almost any microfluidic system. The symmetry of the Fourier modes can be used to extract the stress-strain relationship and to determine viscoelastic cell parameters directly by applying simplest model assumptions. We show that our approach is independent of cellular shape. Using a cell line as well as primary blood cells, we demonstrate that dynamic RT-DC is capable to determine an apparent Young's modulus as well as an apparent viscosity with throughput rates of up to 100 cells per second. Interestingly, this technology allows for a rheological comparison amongst cells in a single measurement of whole blood. In addition, we show a label-free statistical discrimination of major cell types including B- and CD4+ T-lymphocytes based on material properties. The latter has previously only been possible using multi-dimensional machine learning techniques[32,33].

## Results

**High-throughput single cell rheology**. The high-throughput single cell rheology setup is based on the platform of real-time deformability cytometry (RT-DC), which has been published earlier[28]. The main experimental design consists of a microfluidic chip where suspended cells are driven through a central channel and deform due to shear and normal stresses only (Fig. 1a). By using on-the-fly image acquisition and simultaneous analysis of projected surface area $A$ and cell perimeter $P$, RT-DC is capable to calculate a deformation parameter $d$ based on the circularity $c$ of a cell inside a microfluidic channel:

$$d = 1 - c = 1 - \frac{2\sqrt{\pi A}}{P}. \quad (1)$$

An analytical model which has recently been extended numerically allows for extracting an Young's modulus $E$ as a material parameter[30,31]. However, RT-DC is limited to a static region-of-interest (ROI) at the rear part of the microfluidic channel, relies on the assumption of steady-state deformations as well as an initially spherical cell shape, and cannot track the dynamics of shape changes. This would be a prerequisite to derive elastic and viscous material properties of single cells.

Here, we introduce dynamic RT-DC (dRT-DC), a real-time and high-throughput method, to perform single cell rheology of suspended cells on a millisecond timescale. The full field of view of the camera is adjusted to a ROI that covers the entire length of a microfluidic channel, while a moving sub-ROI is used to track the trajectory of cells inside the constriction (Fig. 1a). In its current implementation and by using standard computer hard- and software, dRT-DC is capable to follow more than 100 cells per second in real time. Each single cell trace can consist of more than 100 data points (Fig. 1b and Supplementary Figure 1).

For a cell approaching the channel, we observe an increase in deformation reaching a maximum directly at the inlet (Fig. 1c, top). Interestingly, two distinct dynamic processes are apparent inside the constriction. First, cell deformation approaches a minimum before, second, it increases again and the cell shape adapts a bullet-like steady state given by the Poiseuille flow profile[30]. Both processes can be approximated by an exponential fit function (yellow and green line in Fig. 1c, top) and qualitatively understood considering the hydrodynamic stress distribution inside our microfluidic system. The velocity gradient in flow direction at the inlet and outlet (Supplementary Figure 2a and b) leads to a peak stress $\sigma_{inlet}$ while a smaller stress amplitude $\sigma_{channel}$ is found inside the channel (Fig. 1c, bottom and Supplementary

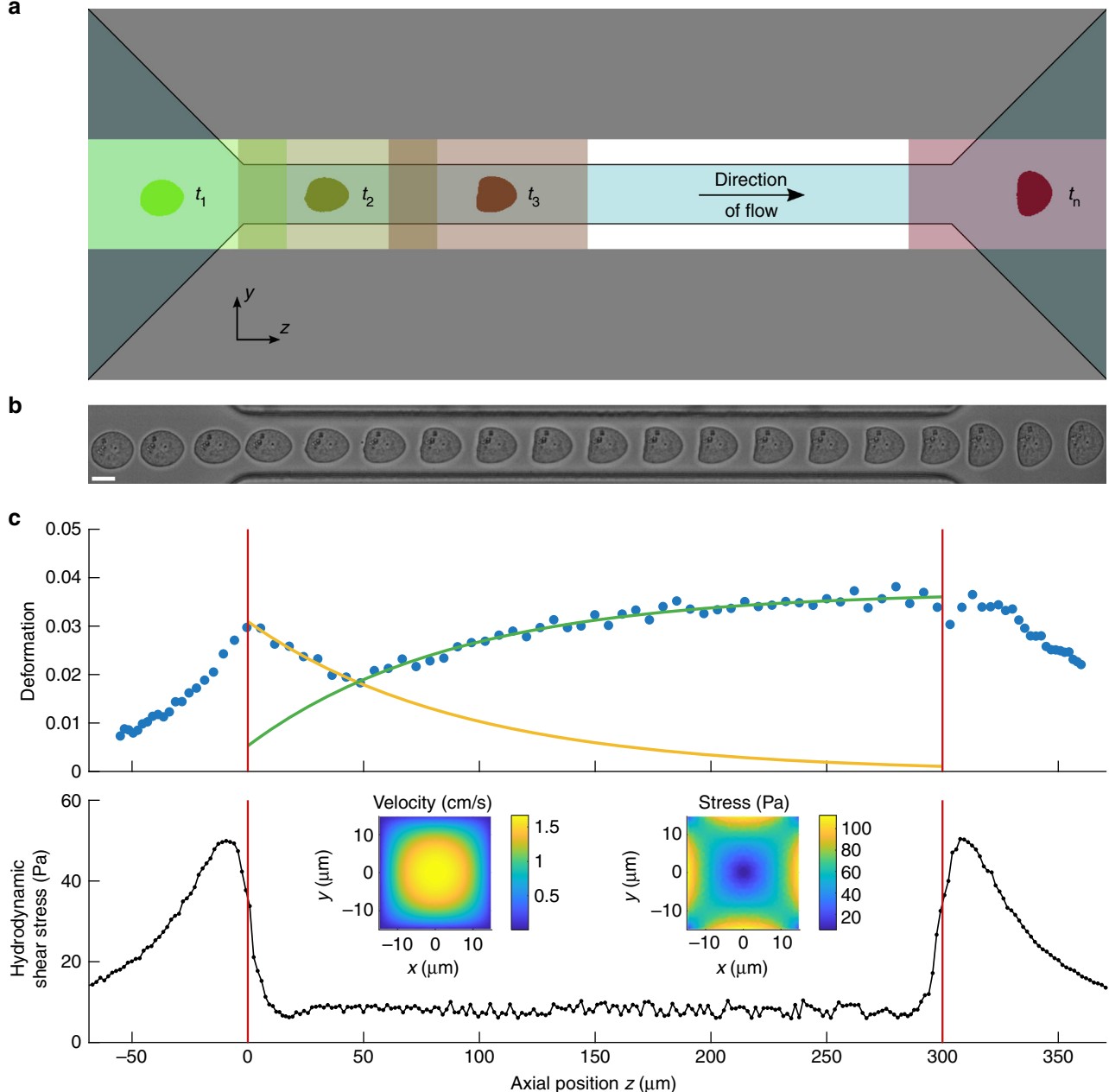

**Fig. 1** Dynamic RT-DC. **a** Sketch of experimental setup showing microfluidic channel with translocating cell and corresponding region-of-interests at selected time points $t_1$, $t_2$, $t_3$, and $t_n$. **b** Time series of a suspended cell while translocating a 30 μm × 30 μm channel of 300 μm length. Length of the scale bar is 10 μm. **c** Deformation (1-circularity) trace of 92 single cell measurements along axial position $z$ in the channel (blue dots). The yellow and green line originate from an exponential fit characterizing two apparent cell relaxation modes (top). The red vertical lines indicate channel inlet and outlet position. Finite element method (FEM) calculations of hydrodynamic shear stress profile along the same microfluidic channel axis at a flow rate of 8 nl s$^{-1}$ in the absence of cells (bottom). Insets show velocity and stress distribution of the channel cross-section

Figure 2c). Apparently, the response of the cell is governed by both, a slow increase in hydrodynamic stress outside the channel towards a maximum at the inlet and a sudden but smaller step stress inside the constriction.

**Shape modes disentangle hydrodynamic stress and cell strain**. Next, we would like to answer the question how the two dynamic processes of decreasing and increasing cell deformation inside the channel can be disentangled and understood quantitatively. This is of fundamental importance to extract material properties from RT-DC data since simple rheological models, e.g., a Kelvin-Voigt

model, require measurements of cell deformation as a response to a well-defined stress or vice-versa[20,34]. Using Fourier analysis, we determined the principal modes of each cell shape from its contour (Fig. 2a and Supplementary Figure 3a) and calculated the first ten Fourier coefficients at each position along the trajectory of the cell inside our microfluidic system (Fig. 2b and Supplementary Figure 3b).

Shape mode analysis was performed for more than 1000 HL60 cells and the results were normalized to the first Fourier coefficient $a_0$ representing the cell radius. Observing the mean (yellow line in Fig. 2b) over channel position reveals a maximum at the inlet (left red line in Fig. 2b and

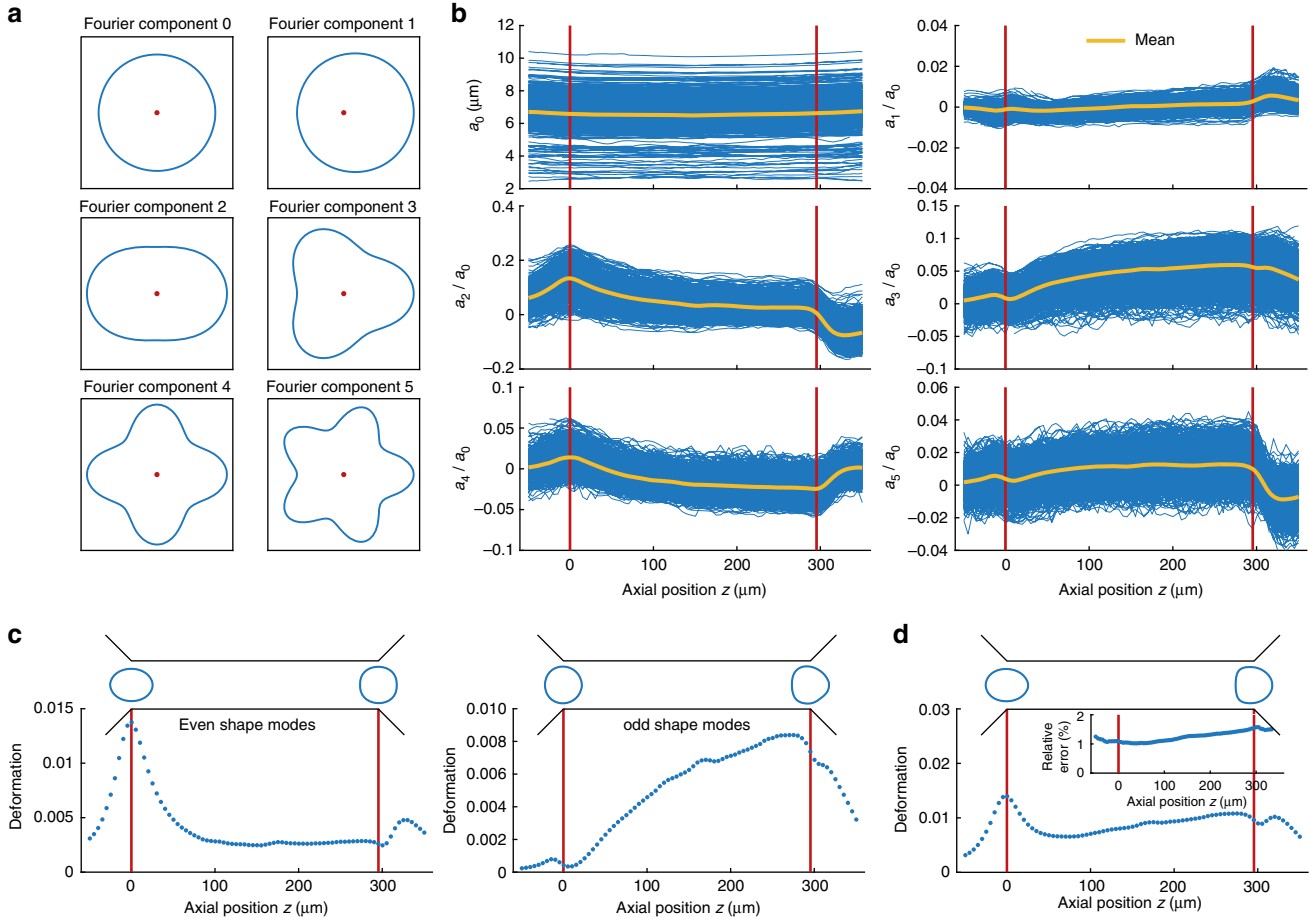

**Fig. 2** Fourier decomposition of cell shape. **a** First six components of radial Fourier function (shape modes) including component zero, red dots indicate the origin. **b** Shape mode analysis applied to dynamic measurements of HL60 cells arranged for even and odd coefficients. Each blue trace represents the amplitude of a Fourier component of a single cell over the axial position $z$ in the channel ($n = 1580$ cells). The yellow lines indicate the mean over all traces whereas the red vertical lines visualize the inlet and outlet position. **c** Reconstruction of cell deformation from first four even (left panel) and first five odd (right panel) Fourier coefficients and $a_0$ with mean cell shapes at channel inlet and outlet (top). **d** Reconstruction of cell deformation from first ten Fourier coefficients, including mean cell shapes (top). Inset shows the relative error in circularity of the reconstructed traces relative to the original dataset. Measurements have been carried out in a $30 \times 30\,\mu m$ channel at a flow rate of $8\,nl\,s^{-1}$

Supplementary Figure 3b) for all even Fourier coefficients followed by a relaxation into a steady state while odd coefficients increase from zero amplitude to a maximum at the end of the channel (right red line in Fig. 2b and Supplementary Figure 3b). These dynamics can be understood when transforming both, the even ($a_2$, $a_4$, $a_6$ and $a_8$) including ($a_0$), and the odd coefficients ($a_1$, $a_3$, $a_5$, $a_7$ and $a_9$) including ($a_0$) respectively, back into the deformation measure introduced in Equation 1. Of note, Fourier coefficients $b_k$ represent the angular orientation of a cell inside the channel revealing a mean value close to zero and are of importance for single cells which are slightly displaced from the channel center (Supplementary Figure 3c).

For the even Fourier coefficients we find a peak in the reconstructed deformation trace $d_{even}$ at the channel inlet followed by a relaxation into a steady state close to zero deformation (left panel in Fig. 2c). This behavior originates from the fluid velocity gradient in flow direction (Supplementary Figure 2b) yielding a maximum in hydrodynamic stress at the inlet (Fig. 1c and Supplementary Figure 2c). The corresponding contour of an elongated cell possesses the same symmetry of shape changes along the channel axis as the even shape modes ($a_0$), $a_2 \ldots a_8$ (Fig. 2a and left panel in Fig. 2c). In contrast, a deformation trace reconstructed from only the first five odd

Fourier coefficients displays a continuous increase from $d_{odd} \approx 0$ at the inlet to $d_{odd} = max.$ at the outlet. The characteristic bullet contour reveals the same symmetry of shape changes as the odd shape modes ($a_0$), $a_1 \ldots a_9$ (Fig. 2a and right panel in Fig. 2c). Reconstruction of the complete trace from the first ten Fourier components (Fig. 2d) resembles the original data while the relative error in deformation magnitude is around 1% or slightly above (Fig. 2d, inset).

In summary, shape mode analysis allows to disentangle the cell response to the inlet peak stress $\sigma_{inlet}$ from the channel stress $\sigma_{channel}$ (Fig. 1c). This implies that $d_{even}$ can be seen as the deformation trace $d_{inlet}$ representing shape dynamics caused by $\sigma_{inlet}$ while $d_{odd}$ can be understood as the deformation trace $d_{channel}$ due to the constant stress $\sigma_{channel}$ (Fig. 2c). The latter can be approximated by a step-function and is given by the hydrodynamic stress around the cell inside the constriction (Supplementary Figure 4). Applying an analytical model published earlier $d_{channel}$ enables to establish a stress-strain relationship[30]. Therewith, rheological parameters can directly be determined from the channel data representing a creep-compliance experiment[35].

**Single-cell rheology on cell line and primary cells**. For performing rheology on single cells we extract four parameters from

dRT-DC data after Fourier decomposition and reconstructing the deformation versus time trace from even and odd Fourier coefficients separately. The cell response to the peak stress at the channel inlet $d_{inlet}$ is reconstructed from the first four even shape modes and $a_0$ representing the cell radius. For this inlet deformation trace, the local maximum defines the peak deformation $\hat{d}_{inlet}$ while the relaxation into a steady state is quantified by the corresponding time constant $\tau_{inlet}$ (Fig. 3a, top). The response of the cell $d_{channel}$ to the constant stress $\sigma_{channel}$ inside the constriction is reconstructed from $a_0$ and the first five odd shape modes. Reaching a maximum at the channel outlet, $d_{channel}$ is described by the steady-state deformation $\hat{d}_{channel}$ relative to the inlet and the characteristic time constant $\tau_{channel}$ (Fig. 3a, bottom). Both timescales, $\tau_{inlet}$ and $\tau_{channel}$, originate from an exponential fit to our reconstructed data traces (yellow lines in Fig. 3a) assuming a linear cell response (Methods).

In a first set of experiments HL60 cells have been used as a model system and we studied the sensitivity of dRT-DC towards cytoskeletal alterations. Treatment with 1 µM cytochalasin D (CytoD), which depolymerizes filamentous actin, leads to an increase in $\tau_{inlet}$ and $\tau_{channel}$ (Fig. 3b) as well as in both deformation measures $\hat{d}_{inlet}$ and $\hat{d}_{channel}$ (Fig. 3c) relative to the wildtype and the corresponding DMSO control (0.25% (v/v)). Between the latter two no differences are observed.

A statistical analysis over three biological replicates each consisting of ~1000 single cell rheological experiments or more was carried out using linear mixed models (see Methods) and confirms the observed trend. Both, $\tau_{inlet}$ and $\tau_{channel}$, are significantly increased relative to the DMSO control as well as to the wildtype (Fig. 3d). Also, the deformation measures $\hat{d}_{inlet}$ and $\hat{d}_{channel}$ rise significantly when exposing HL60 cells to 1 µM CytoD (Fig. 3e). A direct comparison between the DMSO sample and the wild-type reveals no effect on $\tau_{inlet}$ and $\tau_{channel}$ as well as on $\hat{d}_{inlet}$ and $\hat{d}_{channel}$ (Fig. 3d, e).

Next, we use the channel deformation trace to extract rheological parameters from our data. For traces of $d_{channel}$ revealing a steady state (Fig. 3a, bottom), an analytical model can be used to obtain an apparent Young's modulus $E$[30,31] and an apparent viscosity $\eta$ assuming linear viscoelasticity (see Methods)[20]. Briefly, finite element method (FEM) simulations are carried out to obtain the hydrodynamic surface stress distribution of a cell moving in a microfluidic channel (Supplementary Figure 4). For a cell moving within a $30 \times 30$ µm channel at a flow rate of 8 nl/s a mean surface shear stress of 142 Pa (dashed line in Fig. 3a, bottom) is determined. Using this stress distribution the cell deformation can be predicted and mapped to an apparent Young's modulus[30].

Applying our analytical model to biological replicates, HL60 cells exposed to 1 µM CytoD are found with $E = 0.28 \pm 0.03$ kPa, which is significantly reduced compared to the wildtype and DMSO control with an apparent Young's modulus of $E = 0.40 \pm 0.02$ kPa and $E = 0.43 \pm 0.01$ kPa, respectively (Fig. 3f).

The constant stress inside the constriction is not only a requirement of our analytical model to determine $E$, but also enables the interpretation of $\tau_{channel}$ as a cell response to a simple step stress. Under the assumption of a linear response following a Kelvin-Voigt model, the characteristic time $\tau_{channel} = \eta / E$ allows to calculate an apparent cell viscosity[20]. Here, we find a slight increase from $\eta = 3.43 \pm 0.16$ Pa s for the wildtype to $\eta = 3.67 \pm 0.05$ Pa s for the DMSO control, which reveals a significant reduction to $\eta = 2.76 \pm 0.2$ Pa s after CytoD treatment (Fig. 3g).

A re-evaluation of our HL60 wild-type data with respect to power-law rheology and a subsequent comparison to a Kelvin-Voigt model shows a smaller root-mean-square error for fitting an exponential function (Supplementary Figure 5a). This result might be linked to the fact that our time traces extend only over

two orders of magnitude in time[35]. Interestingly, the absolute value in apparent Young's modulus for both models differs by less than 20% — a result which supports our experimental and analysis framework (Supplementary Figure 5b).

Next, we perform single cell rheology experiments on primary cells and assess the potential of dRT-DC for characterization of blood cells from different lineages, including sub-populations. Comparing typical individual traces of erythrocytes, granulocytes, and peripheral blood mononuclear cells (PBMCs) in whole blood, we find qualitative differences in deformation as well as translocation dynamics. While red blood cell rheology is dominated by the peak stress at the inlet of the microfluidic channel, leukocyte dynamics reveal a longer relaxation time in the channel and are thus largely affected by the steady-state parabolic flow profile (Fig. 4a).

Interestingly, dRT-DC allows for the extraction of rheological cell properties independent of cell shape. As simple analytical models require spherical cells for deriving material properties from RT-DC measurements[30,31], non-spherical cells, e.g., erythrocytes and activated cells, could not be characterized. Dynamic RT-DC overcomes this limitation by applying the symmetry of the Fourier components and by reconstructing the deformation trace from only odd shape modes. Here, the initial deformation value by means of circularity is always equal to zero corresponding to a spherical contour before cell response to the hydrodynamic stress (Fig. 4b).

For biological replicates of erythrocytes and purified granulocytes, PBMCs, and lymphocytes from three different donors, we determine the characteristic deformations as well as relaxation times (Supplementary Figure 6) and extract the apparent Young's modulus and the apparent viscosity as described above. Of note, purification of leukocytes (see Methods) has been carried out to obtain a sufficient sample number and to reduce the impact of changing material properties over time, but rheological characterization could be equally well performed on a single cell level in whole blood.

For erythrocytes, we find an apparent Young's modulus of $E = 0.10 \pm 0.01$ kPa while granulocytes ($E = 0.43 \pm 0.05$ kPa) and PBMCs ($E = 0.62 \pm 0.04$ kPa) are significantly stiffer (Fig. 4c). Estimation of the apparent viscosity from the characteristic times $\tau_{channel}$ also yields statistically significant differences between $\eta = 0.24 \pm 0.18$ Pa s for erythrocytes, $\eta = 5.22 \pm 0.34$ Pa s for granulocytes and $\eta = 8.70 \pm 0.45$ Pa s for PBMCs (Fig. 4d).

Dynamic RT-DC also allows for a discrimination of B- and CD4+ T-lymphocytes based on material properties. Interestingly, only minor differences are observed in the channel deformation measure $\hat{d}_{channel}$ (Fig. 4c, inset) and its derived parameters, the apparent Young's modulus (Supplementary Figure 6b top, inset) as well as the apparent viscosity (Supplementary Figure 6b bottom, inset). In contrast, an analysis of the inlet deformation trace reveals a significant increase in $\hat{d}_{inlet}$ for CD4+ T-cells (Fig. 4d, inset). In fact, lymphocytes are the only blood cells included in this study where sub-populations can better be identified based on the inlet dynamics.

Finally, we use a logistic model to assess the capability of dRT-DC to discriminate leukocytes on a single cell level. Following a multivariate approach incorporating cell size $A$, apparent Young's modulus $E$, and apparent viscosity $\eta$ the Akaike information criterion (AIC) is used for model selection (see Methods). Comparing granulocytes and peripheral blood mononucleated cells we find a two-parametric model of $A$ and $\eta$ having the lowest AIC, which identifies PBMCs with a sensitivity exceeding 80% and an area under the curve AUC = 0.88 (Supplementary Figure 7a and Supplementary Table 1). For classification of CD4+ T cells we have been analyzing logistic models of cell size

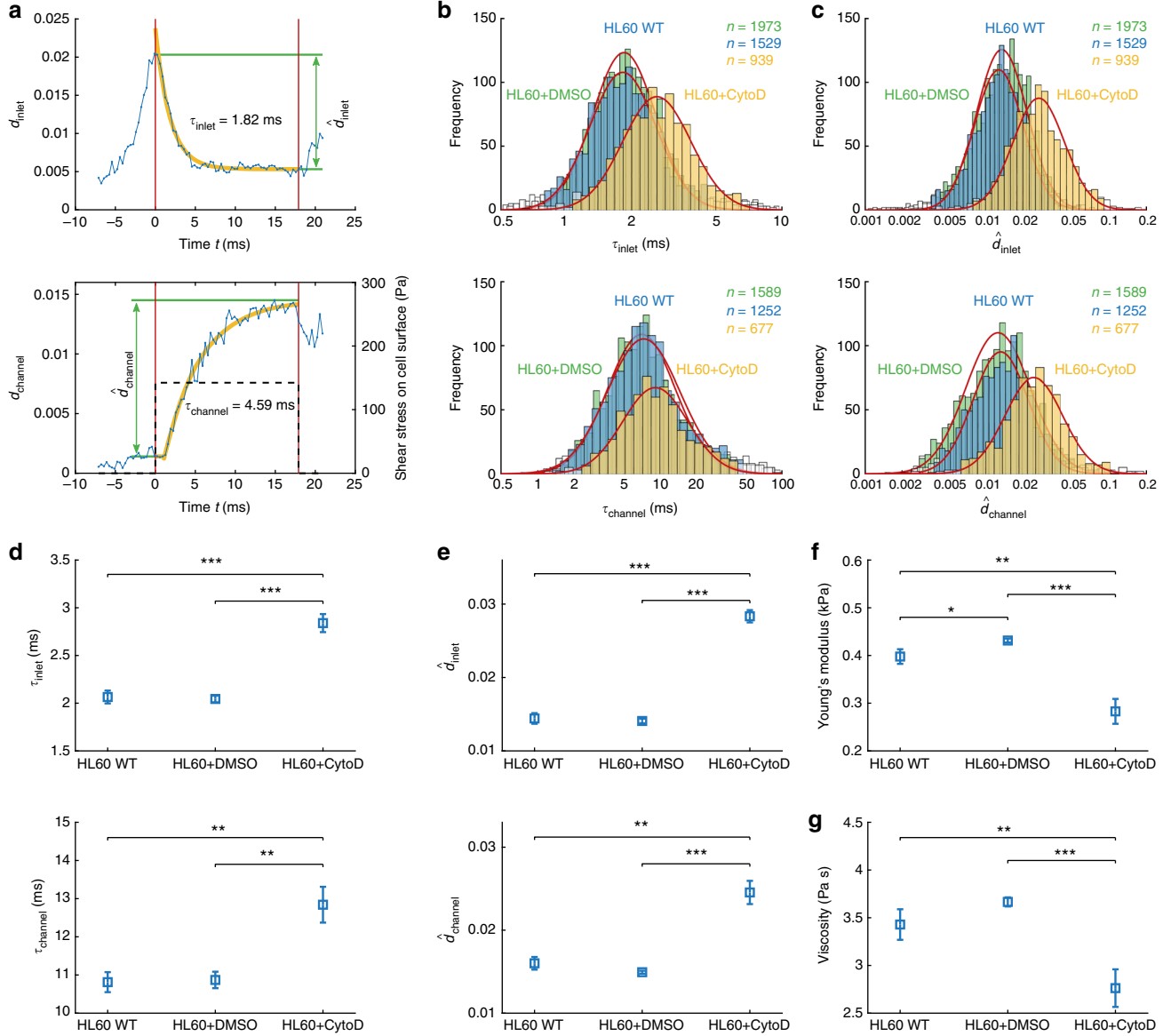

**Fig. 3** Single-cell rheology on HL60 cells. **a** Reconstructed deformation trace $d_{inlet}$ (top) and $d_{channel}$ (bottom) from first four even (top) and first five odd (bottom) cell shape modes including $a_0$, respectively (blue). Graphs show relaxation times $\tau_{inlet}$ and $\tau_{channel}$ (yellow) as well as peak deformations $\hat{d}_{inlet}$ and $\hat{d}_{channel}$ (green). Dashed line (bottom) represents mean shear stress on cell surface. The red vertical lines indicate channel inlet and outlet position. **b** Histograms of characteristic times $\tau_{inlet}$ and $\tau_{channel}$ are binned logarithmically and show wild-type cells (blue), cells after treatment with 1 μM CytoD (yellow) and the corresponding DMSO control (0.25% (v/v), green). **c** Histograms of peak deformation $\hat{d}_{inlet}$ and $\hat{d}_{channel}$ are binned logarithmically and show wild-type cells (blue), cells after treatment with 1 μM CytoD (yellow) and the corresponding DMSO control (green). In **b** and **c** data in a confidence interval of 3σ have been taken into account for the log-normal fit (red line), others are considered as outliers (white bins). **d-g** Statistical analysis for three biological replicates comparing wild-type cells ($n = 3382$), cells after treatment with 1 μM CytoD ($n = 2643$) and corresponding DMSO control ($n = 3965$) for characteristic timescales $\tau_{inlet}$ and $\tau_{channel}$ **d**, for peak deformation $\hat{d}_{inlet}$ and $\hat{d}_{channel}$ **e**, for Young's modulus $E$ **f** and viscosity $\eta$ **g**. Measurements have been carried out in a 30 × 30 μm channel at a flow rate of 8 nl s$^{-1}$. The mean shear rate of 5100 s$^{-1}$ and the mean shear stress of 142 Pa on the cell surface has been derived from finite element method simulations considering the full microfluidic geometry. Statistical significance has been calculated from linear mixed models and error bars represent standard error of the mean (*$p < 0.05$; **$p < 0.01$; ***$p < 0.001$)

$A$, the peak deformation $\hat{d}_{inlet}$ and the inlet relaxation time $\tau_{inlet}$. Here, a two-parametric model of $\hat{d}_{inlet}$ and $\tau_{inlet}$ reveals the lowest AIC with a sensitivity of 76% and an AUC = 0.62 (Supplementary Figure 7b and Supplementary Table 2).

## Discussion

Rheological cell properties provide essential insights into biological functions regarding health and disease. Being label free, this biomarker allows for an unbiased phenotyping of cells as well as cellular sub-populations and is complementary to established assays in molecular biology. The latter always require an apriori knowledge about the molecular target of interest.

Within the existing technologies for rheological cell phenotyping, dynamic RT-DC fulfills the unmet requirement for high-throughput real-time analysis of heterogeneous samples with accompanying derivation of material properties from simplest model assumptions while being independent of cell shape.

The decomposition of cell shape into Fourier modes reveals two dynamic processes for a suspended cell passing a microfluidic

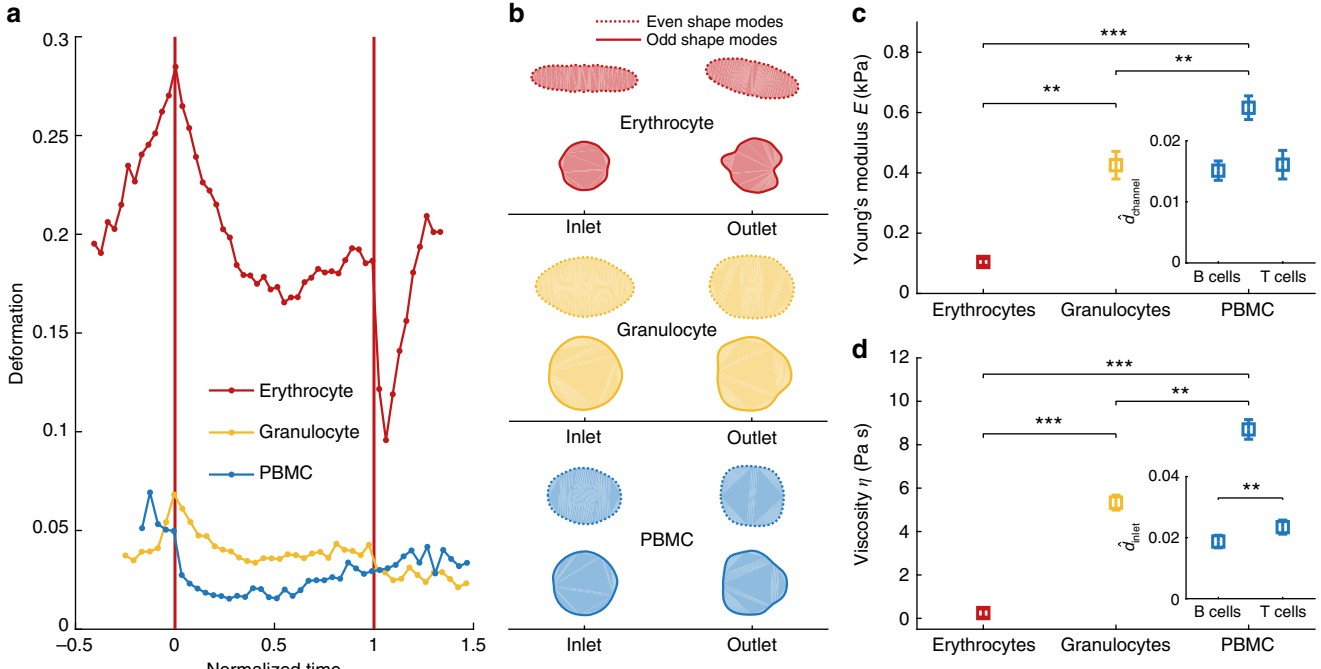

**Fig. 4** Single cell rheology on peripheral blood cells. **a** Dynamic RT-DC traces of individual cells in whole-blood showing erythrocytes (red), granulocytes (yellow) and peripheral blood mononuclear cells (PBMC, blue). Translocation time has been normalized to account for different velocities. **b** Cell shapes of typical blood cells reconstructed from even (dashed line) and odd (solid line) shape modes showing erythrocytes (red), granulocytes (yellow) and PBMCs (blue). **c** Apparent Young's modulus $E$ and **d** apparent viscosity $\eta$ of purified cell samples. Insets show a comparison of the peak channel deformation $\hat{d}_{channel}$ **c** and the peak deformation $\hat{d}_{inlet}$ **d** for B cells and CD4+ T cells at the inlet. Statistical analysis in **c** and **d** includes data of three biological replicates from $n = 381$ erythrocytes, $n = 243$ granulocytes, $n = 130$ PBMCs, $n = 737$ B cells and $n = 3079$ CD4+ T cells. Experiments have been carried out in a 20 × 20 μm channel at a flow rate of 4 nl s⁻¹. The mean shear rate of 9700 s⁻¹ and a mean stress of 216 Pa on the cell surface (granulocytes and PBMCs) as well as the mean shear rate of 8600 s⁻¹ and a mean stress of 128 Pa on cell surface (erythrocytes) has been derived from finite element method simulations considering the full microfluidic geometry. Statistical significance has been calculated from linear mixed models and error bars represent standard error of the mean (**$p < 0.01$; ***$p < 0.001$)

channel. Even Fourier modes reflect a symmetry parallel as well as perpendicular to the channel axis and cell shapes reconstructed only from these components describe the cell response to the time-dependent flow profile of the inlet region. In contrast, odd Fourier modes are sensitive to shape changes perpendicular to the channel axis, only. Reconstruction of cell shape from these components reveals a dynamic process of zero deformation at the channel inlet and maximum deformation at the channel outlet as a response to the constant shear flow inside the constriction. Since this separation between even and odd Fourier modes is entirely based on symmetry arguments, dynamic RT-DC extends its parameter space to two deformation measures and two characteristic times within one experimental assay.

For a given cell type, these parameter pairs reveal similar experimental magnitudes supporting the model assumption of a linear homogeneous material. As a consequence, the deformation reconstructed from the odd shape modes can be considered as the strain response to a constant stress inside the channel. The characteristic time to reach the steady state can be analyzed using a simple Kelvin-Voigt model. It yields, together with an analytical model published earlier[30], an apparent Young's modulus and an apparent viscosity. Remarkably, the application of Fourier decomposition allows for rheological cell characterization fully independent of cell shape, because traces reconstructed from odd Fourier modes always start with zero deformation, which is a requirement of simple mechanical models[31].

We applied our experimental framework to HL60 cells as a model system and determined an apparent Young's modulus of ~0.4 kPa, which is in agreement with previously published results[27], but lower than earlier RT-DC data[30]. This deviation potentially originates from strain-stiffening effects[2] and the feature of dRT-DC that specifically takes into account the initial cell shape and does not rely on ensemble averages to correct for non-spherical objects.

Depolymerization of filamentous actin by CytoD leads to a reduction in the apparent Young's modulus as reported earlier for HL60 and 3T3 cells[27,36]. Although these previous studies successfully captured changes in elasticity by applying power-law rheology no significant differences in the fluidity were observed. Here, using dRT-DC and applying a linear viscoelastic model also allows to link actin depolymerization to a reduction in the apparent viscosity. Interestingly, the apparent viscosity of ~4 Pa s is almost two orders of magnitude lower than previously published, which might be due to shear-thinning effects in the cytoskeleton[37]. In fact, experimental and theoretical studies on the cytoplasmic viscosity of leukocytes revealed a shear-rate dependency following a power-law[38].

Finally, we have performed dRT-DC on peripheral blood and determined viscoelastic properties for erythrocytes, granulocytes, and peripheral blood mononuclear cells within a single measurement. This has not been possible before as state-of-the-art approaches always required purification of the cell type of interest[39,40]. However, sample preparation and cell isolation potentially alter cellular state and function, e.g. by using fluorescent labels. The capability of dRT-DC to identify and characterize cells in heterogeneous samples independent of their shape might be highly relevant to understand immune response during chronic diseases. Here, it has been shown that infections

with mycobacteria, Plasmodium, and hepatitis are known to increase susceptibility to other pathogens, but the underlying processes are largely unknown[41]. A direct and unbiased cell analysis based on single cell rheology within a complex sample as whole blood would therefore provide a complementary approach to study these coinfections.

Previous experiments using RT-DC have already demonstrated a correlation between malaria parasite infection and reduction of host cell membrane bending modulus[8,42]. An extension of this analysis from a specific cell type towards a mechanoprofile of interacting immune cells in whole blood might shed new light on these bystander effects during coinfections and help to elucidate novel infections mechanisms.

In this context, we provide evidence that dRT-DC is capable to discriminate between isolated leukocytes of different lineages including B- and CD4+ T-lymphocytes with statistical significance and a sensitivity of 76%. Therefore, dRT-DC is to the best of our knowledge the only technology that can distinguish between those cells without any external labeling or machine-learning framework. This is of essential importance for cell enrichment based on intrinsic material properties which require real-time operation at high throughput. Moving towards this need, dynamic RT-DC integrates the identification of material properties from simplest model assumptions into on-the-fly high-throughput mechanical phenotyping and bridges the gap between molecular and rheological analysis of cells.

## Methods

**Experimental setup.** The dynamic RT-DC setup is based on the real-time deformability cytometry system[28]. Briefly, a microfluidic chip is assembled on the Accellerator (Zellmechanik Dresden) which consists of an inverted microscope with CMOS camera, a dedicated syringe pump, a microsecond pulsed LED illumination, and a dedicated acquisition software. The microfluidic chip is made out of poly-dimethylsiloxane (PDMS) and glass, has two inlets (one for the sample solution and one for sheath) and one outlet. Inlets and outlet are connected by a central constriction of 300 μm length and a cross-section of 20 μm × 20 μm or 30 μm × 30 μm, respectively.

To enable a continuous high-throughput tracing of dynamic cell changes, the RT-DC acquisition software has been adapted and a new analysis software has been developed. The field of view (1280 × 80 pixels) covers the full length and width of our channel including the inlet and outlet area (Fig. 1a, b) where our camera (MC1362, Mikrotron) takes images at frame rates between 2000 and 6000 frames per second (fps). Detection of a cell is based on image binarization and a border following algorithm as published earlier. Once a cell appears in the inlet region, our algorithm crops a region-of-interest (ROI) of 250 × 80 pixels around the cell, which moves along the channel until the outlet is reached (rectangles in Fig. 1a). Decision of ROI motion is based on the cell center-of-mass. For positions exceeding 70% of ROI length, the ROI moves by 48% ensuring that slow and fast cells can be tracked during their passage through the microfluidic channel. The small size of the ROI ensures that cell tracking is stable and not interfering with other cells in the channel.

In each iteration our algorithm calculates the position of the cell within the ROI and the channel, deformation and cell size. The current optical resolution of our system of 0.34 μm per pixel allows for the analyses of 100 cells per second where each trace consists of at least 14 data points. Cell velocities of up to 24 cm s$^{-1}$ are achievable. The use of a high-viscous cell carrier that enables considerable cell deformation already at low flow rates effectively increases the number of data points per trace (Supplementary Figure 1).

**Shape mode analysis.** Shape mode analysis is based on a custom Matlab R2017a script (Mathworks) where the convex hull contour is extracted from a single cell image. The contour is described by a closed polygon $(x_{cont,n}, y_{cont,n})$ with $n = 0, 1, ..., N-1$ where $N$ represents the number of contour points and $(x_{cont,N}, y_{cont,N}) = (x_{cont,0}, y_{cont,0})$.

In the first step of our algorithm the contour is piecewise linearly interpolated by 49 points between each pair of contour points resulting in the interpolated tuple of vectors $(x_n, y_n)$. Next, each contour is transformed into polar coordinates $(r_n, \varphi_n)$ with respect to its center-of-mass $(x_{com}, y_{com})$:

$$r_n = \sqrt{(x_n - x_{com})^2 + (y_n - y_{com})^2}, \quad \varphi_n = \arctan2(y_n - y_{com}, x_n - x_{com}), \quad (2)$$

with

$$x_{com} = \tfrac{1}{6A} \sum_{n=0}^{N-1} (x_n + x_{n+1})(x_n y_{n+1} - x_{n+1} y_n),$$

$$y_{com} = \tfrac{1}{6A} \sum_{n=0}^{N-1} (y_n + y_{n+1})(x_n y_{n+1} - x_{n+1} y_n),$$

$$A = \tfrac{1}{2} \sum_{n=0}^{N-1} (x_n y_{n+1} - x_{n+1} y_n).$$

The interpretation as an angle dependent radius function $r_n(\varphi_n)$ allows for applying a discrete Fourier transformation (DFT) with Fourier coefficients $a_k$ and $b_k$:

$$a_k = \tfrac{1}{\pi} \sum_{n=0}^{N-1} r_n \cos(k\varphi_n) \Delta\varphi_n, k \geq 0$$
$$b_k = \tfrac{1}{\pi} \sum_{n=0}^{N-1} r_n \sin(k\varphi_n) \Delta\varphi_n, k \geq 1 \qquad (3)$$

with

$$\Delta\varphi_n = \frac{1}{2}(\varphi_n - \varphi_{n-1}) + \frac{1}{2}(\varphi_{n+1} - \varphi_n).$$

In our system $b_k$ represent the angular orientation of the cell inside the channel and their mean magnitude is expected to be zero due to shape symmetry in the direction of flow. In experiments $b_k$ are statistically distributed with an expectation value close to zero (Supplementary Figure 3c). For cell contour reconstruction, the first ten Fourier coefficients $a_k$ ($k \leq 9$) are sufficient. Whereas $a_0$ is a measure of cell size, $a_1$ gives eccentricity, and $a_2$ represents ellipticity. Higher shape modes describe shapes near to a $s$-pointed star, where $s$ is the order of the Fourier coefficient (Fig. 2a and Supplementary Figure 3a). A size-invariant representation of the contour is achieved by a normalization of all Fourier coefficients to $a_0$. To distinguish between the peak stress response at the inlet and the response to the steady-state stress inside the channel, the two subsets of shapes are reconstructed separately, $r_{rec,even}(\varphi_{rec})$ for even coefficients and $r_{rec,odd}(\varphi_{rec})$ for odd coefficients:

$$r_{rec,even}(\varphi_{rec}) = \frac{a_0}{2} + \sum_{k=1}^{4} [a_{2k}\cos(2k\varphi_{rec}) + b_{2k}\sin(2k\varphi_{rec})],$$

$$r_{rec,odd}(\varphi_{rec}) = \frac{a_0}{2} + \sum_{k=0}^{4} [a_{2k+1}\cos((2k+1)\varphi_{rec}) + b_{2k+1}\sin((2k+1)\varphi_{rec})] \qquad (4)$$

From the reconstructed cell shape, deformations can be derived using Equation 1, resulting in $d_{inlet}$ and $d_{channel}$.

**Cell culture.** *HL60 cells*, a myeloid precursor cell line, is used as a model system for suspended cells (courtesy of Dan and Ada Olins). Cells are cultured in RPMI-1640 medium (BioWest) with 10% FCS (Gibco), 1% penicillin/streptomycin (BioWest) and 2 mM L-Glutamin (BioWest) using a standard incubator at 37 °C, 5% CO$_2$ and 95% air. Every 48 h cells are centrifuged at 200 rcf for 5 min. (Allegra X-15R, Beckman Coulter), the supernatant discarded and re-suspended to a concentration of approximately $1.5 \times 10^5$ cells per milliliter. All experiments have been carried out during log phase, ~36 h after splitting. Viability of cells has been assessed to ~95% using Trypan Blue. Cells have been checked for Mycoplasma infection.

*Peripheral blood cells* have been obtained from healthy blood donors with written consent and under approval of the Universitätsmedizin Greifswald. Blood samples are withdrawn with a 20-gauge multifly needle into a S-monovette of 6 ml sodium citrate by vacuum aspiration (BD). In whole blood measurements, blood is diluted 1:18 in phosphate buffered saline (PBS−/−, without Ca$^{2+}$/Mg$^{2+}$, BioWest) being supplemented with 1% (w/v) methylcellulose (Sigma-Aldrich). For measurements of erythrocytes, blood is diluted 1:180 in Cell Carrier B (PBS−/− and 0.6% methylcellulose, Zellmechanik Dresden) to reduce cell concentration and shear stress.

*Granulocyte isolation* has been performed using the EasySep Direct Human Pan-Granulocyte Isolation Kit (StemCell). Buffy coats from whole-blood donations of healthy blood donors (Department for Transfusion Medicine, Universitätsmedizin Greifswald) are mixed 1:1 with PBS−/− before negative magnetic bead isolation according to supplier instruction. Following a final centrifugation at 200 rcf for 5 min (Allegra X-15R, Beckman Coulter), cells are re-suspended in 100 μl PBS−/− and 1% (w/v) methylcellulose to a final concentration of approximately $0.5 \times 10^7$ cells per milliliter. Purity of granulocytes (91.8%) has been confirmed using flow cytometry (LSRII, BD, Supplementary Figure 8a).

*Peripheral blood mononuclear cells* (PBMCs) have been isolated using Ficoll gradient centrifugation. Briefly, whole blood from buffy coats (Department for Transfusion Medicine, Universitätsmedizin Greifswald) is mixed 1:1 with PBS−/−. Next, 10 ml of Ficoll (GE Healthcare) in 50 ml falcon tubes is carefully overlaid with 30 ml of buffy coats/PBS−/− mixture. After an initial centrifugation at 400 rcf for 40 min. (no breaks, Allegra X-15R, Beckman Coulter), the interphase is removed and transferred into a new 50 ml falcon tube. In the following, 50 ml of PBS−/− are added and the suspension is centrifuged at 300 rcf for 10 min. In two

iterations, the supernatant is discarded, the pellet is re-suspended in 50 ml PBS−/− and centrifuged at 200 rcf for 10 min. PBMCs are re-suspended in PBS−/− and 1% (w/v) methylcellulose to a final concentration of approximately $0.5 \times 10^7$ cells per milliliter.

Isolation of T- and B-lymphocytes has been carried out from PBMCs using the EasySep Human CD4+ T-cell and B-cell enrichment kit (StemCell), respectively. Briefly, isolated PBMCs are adjusted to a final concentration of $5 \times 10^7$ cells per milliliter in PBS−/− and isolation for T-lymphocytes and B-lymphocytes has been performed according to supplier instruction. After a final centrifugation at 200 rcf for 5 min. cells are re-suspended in 100 µl PBS−/− supplemented with 1% (w/v) methylcellulose to a final concentration of $0.5 \times 10^7$ cells per milliliter. For reducing systematic measurement error from time-dependent deformation changes, purification of B- and T-lymphocytes has been carried out sequentially and the measurement sequence has been altered between experimental replicates. Purity of T-lymphocytes (99.5%) and B-lymphocytes (99.9%) has been confirmed using flow cytometry (LSRII, BD, Supplementary Figure 8b and 8c).

**Drug treatment of HL60 cells.** Inhibition of polymerization of filamentous actin has been done using cytochalasin D (Sigma-Aldrich) capping the plus end of the filaments. Cells are aliquoted at a concentration of $1 \times 10^6$ cells per milliliter and cytochalasin D (CytoD) dissolved in dimethylsulfoxid (DMSO) is added to a final concentration of 1 µM[43]. After an incubation at 37 °C for 10 min cells are centrifuged at 200 rcf for 5 min and re-suspended in PBS−/− and 1% (w/v) methylcellulose as a measurement medium. Control measurements have been carried out using the wild-type cells and the corresponding DMSO control at 0.25% (v/v) concentration.

**Dynamic RT-DC measurement procedure.** Prior to measurement, polymer tubing which connects the syringe pump and the microfluidic chip is flushed extensively with de-ionized water and ethanol. Next, 1 ml syringes (1 ml Luer-Lock syringe, BD) are filled with a measurement buffer, connected to the polymer tubing and the entire system is equilibrated at a flow rate of 100 nl s$^{-1}$ for several minutes. As measurement buffer PBS−/− with 0.6% (w/v) for erythrocytes and 1% (w/v) methylcellulose for all other cells has been used. Addition of methylcellulose increases hydrodynamic stress at low flow rates, thus leading to an improved temporal and spatial resolution at a given frame rate.

In a typical experiment 100 µl of cell suspension is drawn into the tubing, which is then connected to the microfluidic chip. After first cells arrive at central constriction in the field of view, flow rate is reduced to 4 nl s$^{-1}$ and the system is again equilibrated for 10 min. Dynamic RT-DC is carried out at flow rates of 4 nl s$^{-1}$ and 8 nl s$^{-1}$, respectively. One measurement at a given flow rate is usually composed of several thousand data points. While one flow rate is sufficient for full rheological analysis of our samples, initial characterization is usually done for multiple conditions. For experimental comparison, we rescale the flow rate to the average cell surface stress. The stress is obtained from FEM calculations of steady-state cells in our microfluidic channel applying the shear-rate dependency of the measurement buffer viscosity (Supplementary Figure 9).

**Hydrodynamic simulations.** For numerical simulations, we use the standard as well as the CFD packages of COMSOL Multiphysics 5.3a. Neglecting inertial, turbulent, and gravitational terms, the steady-state Navier-Stokes equation simplifies to Stokes flow.

The fluid is set to be incompressible and to the measured density of 1065 kg m$^{-3}$ (DMA4500, Anton Paar). The shear-rate-dependent viscosities $\eta_{1/2}(\dot{\gamma})$ of our measurement buffers have been characterized using a rheometer (MCR502, Anton Paar) and then modeled for PBS−/− with 0.6% (w/v) methylcellulose by a simple power law to:

$$\eta_1(\dot{\gamma}) = K_1 \cdot \left(\frac{\dot{\gamma}}{\dot{\gamma}_0}\right)^{(n_1-1)}, \tag{5}$$

where $K_1$ is a consistency coefficient with $K_1 = 0.16$ Pa s and $n_1$ is the flow behavior index with $n_1 = 0.74$.

For PBS−/− with 1% (w/v) methylcellulose we find:

$$\eta_2(\dot{\gamma}) = K_2 \cdot \left(\frac{\dot{\gamma}}{\dot{\gamma}_0}\right)^{(n_2-1)}, \tag{6}$$

with $K_2 = 0.60$ Pa s and $n = 0.64$. In both equations $\dot{\gamma}$ denotes the hydrodynamic shear rate and $\dot{\gamma}_0 = 1$ s$^{-1}$ (Supplementary Figure 9). This shear-thinning behavior is expected from a polymer solution of methylcellulose and has been reported before[44,45]. In our microfluidic system, viscosities in the order of 10 mPa s have been determined from the relevant shear rate regime.

The simulations are carried out on physics-controlled meshes of an extra fine element size defined by the COMSOL backend. Parametric and material sweeps are performed for numerous conditions with respect to sample and sheath flow, channel geometry and object sizes (Supplementary Table 3). The layout of the microfluidic chips has been reconstructed, and one quadrant simulated, thus, taking advantage of the two symmetry axes through the center line of the design.

Walls have been simulated using the no-slip boundary condition, and laminar flows entering the inlets of sample and sheath were assigned specific rates under constant pressure conditions. At the outlet, backflow is suppressed, reflecting the experimental conditions.

To estimate the shear stress on a cell's surface, a sphere has been modeled inside a 100 µm long channel of 20 µm × 20 µm, respectively 30 µm × 30 µm cross-section, as previously described[30]. The boundary conditions of the channel have been chosen to balance net normal pressure and hydrodynamic shear stress on the sphere's surface (Supplementary Figure 4) within numerical precision.

**Analytical model.** Cell response to the peak stress at the channel inlet is described by the deformation trace $\hat{d}_{inlet}$ from even Fourier modes while cell response to the constant stress inside the channel is illustrated by the deformation trace $d_{channel}$ reconstructed from odd Fourier modes. Using a custom Matlab R2017a script (Mathworks), the characteristic time for reaching the steady state is found by fitting the following exponential function to each trace:

$$d(t) = d_0 + \hat{d}e^{-\frac{t}{\tau}}, \tag{7}$$

where $d(t)$ is the deformation given by cell circularity (Equation 1), $d_0$ is a free fit parameter characterizing the steady-state deformation after relaxation of all time-dependent processes, $\hat{d}$ is the deformation amplitude with $\hat{d}_{inlet} = \hat{d}$ and $\hat{d}_{channel} = -\hat{d}$ and $\tau$ is the characteristic relaxation time $\tau_{inlet}$ or $\tau_{channel}$, respectively.

Quality of the fit is determined by the coefficient of correlation $r^2$ which is defined by:

$$r^2 = 1 - \frac{SS_{res}}{SS_{tot}} = 1 - \frac{\sum_i (y_i - f_i)^2}{\sum_i (y_i - \bar{y})^2}, \tag{8}$$

where $y_i$ represents the data points, $f_i$ the function values of the fit and $\bar{y}$ the mean of datapoints. For all data we use a cutoff of $r^2 \geq 0.6$ (Supplementary Figure 10).

For calculation of cell rheological properties, we use the channel deformation trace $d_{channel}$ only. Here, cells reach a steady-state deformation $\hat{d}_{channel}$, which is defined by the balance of normal and shear forces inside the constriction. In this steady state we derive the cellular surface stress using FEM simulations as described above. The apparent Young's modulus $E$ is then calculated using an analytical model published earlier[30]. Briefly, by coupling the hydrodynamic stress distribution in the channel to linear elasticity theory the deformation of a suspended spherical object can be predicted. Carrying out this analysis for a range of cell sizes and Young's modulus allows for generation of a look-up table, where experimental and predicted deformations can be compared. The stress distribution on the cell surface is derived from flow rate, channel diameter and viscosity for the mean size of each cell type presented in this study including flow-rate dependency and accompanying shear-thinning behavior of the surrounding medium.

While $d_{channel}$ does not represent a simple strain function, e.g. $\varepsilon = \Delta l/l$, due to the parabolic hydrodynamic stress distribution, it is sufficient to represent the dynamic response of the cell to the step stress inside the channel and to derive a relaxation time from the fit of the channel deformation trace $d_{channel}$ to Equation 7.

This relaxation time $\tau_{channel}$ can be considered the characteristic time for the creep-compliance response of the cell to a constant stress $\sigma_{channel}$ (dashed line in Fig. 3a, bottom) inside the channel. Here, an apparent viscosity $\eta$ can be calculated using a simple Kelvin-Voigt model with:

$$\tau_{channel} = \frac{\eta}{E}, \tag{9}$$

where $E$ is the apparent Young's modulus.

Application of these simple models to calculate the Young's modulus and viscosity requires an alignment of the cells in the center of the channel[30]. We verified this condition by analyzing the apparent Young's modulus and the apparent viscosity as a function of the cell displacement from the channel center and find a standard deviation $\sigma < 1$ µm for all conditions. Using a Spearman rank correlation test we find correlation coefficients of $\rho < 0.09$ for all conditions which confirms that there is no correlation between the rheological parameters and lateral cell position (Supplementary Figure 11).

**Calibration of dynamic RT-DC.** Calibration of dRT-DC has been carried out using standardized elastic poly-acrylamide microgel beads from ref. [46]. The rheology of beads with a mean diameter of 15.2 ± 0.8 µm and an elastic modulus of 1.5 ± 0.5 kPa has been characterized in 20 µm × 20 µm and 30 µm × 30 µm channels at a flow rate of 16 nl s$^{-1}$ and 128 nl s$^{-1}$, respectively. Using the peak channel deformation $\hat{d}_{channel}$ and applying our analytical model we obtain an apparent Young's modulus of 2.20 ± 0.03 kPa and 2.00 ± 0.04 kPa (Supplementary Figure 12). The slight deviation from the calibration standard is still within or close to the standard error of the mean and could be attributed to batch-to-batch variation as discussed in ref. [46]. Interestingly, the authors from Girardo et al. also performed measurements on the relaxation time using ensemble averages over hundreds of beads. They find a

value of $0.12 \pm 0.02$ ms, which is remarkably close to $0.36 \pm 0.01$ ms ($20\,\mu m \times 20\,\mu m$ channel) and $0.12 \pm 0.01$ ms ($30\,\mu m \times 30\,\mu m$ channel) for single beads.

**Power-law rheology**. A comparison between Kelvin-Voigt and power-law rheology was performed by fitting the following creep function to dRT-DC data

$$J(t) = \frac{1}{E} \cdot \left(\frac{t}{t_0}\right)^{\beta}, \tag{10}$$

where $J(t)$ is the creep compliance defined by the ratio of the time-dependent strain $\varepsilon_{\text{channel}}$ and the stress $\sigma_{\text{channel}}$ inside the channel, $E$ is the Young's modulus and $\beta$ is the fluidity (Supplementary Figure 5). A reference time of $t_0 = 1$ s was used. For approximating the cellular strain $\varepsilon_{\text{channel}}$ inside the channel, the ratio between maximal cell surface displacement and radius was estimated to 10% and used to derive the creep compliance.

**Statistical data analysis**. Statistical significance has been calculated using linear mixed models (LMM) on three experimental replicates carried out on separate measurement days or using biological replicates[47]. For LMM, a pair-wise comparison is performed where the observed change in an observable is split into a fixed effect, a random effect and random fluctuations. While the random effect and random fluctuations account for systematic and random measurement bias, e.g., change in temperature or donor to donor variability, the fixed effect describes the statistical difference between two groups, e.g., control and compound treatment. Two models, one with $L_{\text{Model}}$ and one without the fixed effect term $L_{\text{NullModel}}$, are fit to the data and the maximum likelihood is calculated. The likelihood ratio indicates how well each model fits the data and the corresponding $p$-value is computed by the Wilks theorem and by using the software ShapeOut (Zellmechanik Dresden).

The predictive potential of dRT-DC has been estimated based on the parameters: cell size $A$, apparent Young's modulus $E$, apparent viscosity $\eta$, the peak inlet deformation $\hat{d}_{\text{inlet}}$ and inlet relaxation time $\tau_{\text{inlet}}$ using logistic regression models. By considering all possible parameter combinations of $A$, $E$, and $\eta$ for the classification of granulocytes and PBMCs and all combinations of $A$, $\hat{d}_{\text{inlet}}$ and $\tau_{\text{inlet}}$ for the classification of B- and CD4+ T-cells model selection is based on Akaike's information criterion (AIC)[48]. For models describing the discrimination between (a) granulocytes and PBMCs as well as between (b) B- and CD4+ T-cells AICs are reported in Supplementary Material (Supplementary Table 1 and 2). The model with the lowest AIC is considered the best model while preference is given to the model with the lower number of parameters if the differences in AIC is less than two. Receiver operating characteristics (ROC) including the area under the ROC curve (AUC) are derived for each model and optimal cutoff values for sensitivity and specificity have been calculated for the intersection in the ROC curve where sensitivity equals specificity[49]. Data are always presented for the model yielding the lowest AIC.

**Code availability**. The source code for data acquisition (LabView 2016) and analysis (COMSOL Multiphysics 5.3a and Matlab R2017a) is available from the corresponding author upon reasonable request.

## Data availability
The datasets generated in this study are available from the corresponding author upon reasonable request. The file format of the raw data is TDMS which can be read and analysed with Matlab or ShapeOut.

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

## Acknowledgements

We thank Andreas Greinacher, Volkmar Liebscher and Ricardo Pires for valuable discussions. We gratefully acknowledge support from the German Federal Ministry of Education and Research (ZIK grant to O.O. under grant agreement 03Z22CN11) and the German Research Foundation (project number 374031971 – TRR240).

## Author contributions

O.O. conceived the project. B.F. developed the experimental setup. F.C. performed FEM simulations and rheological characterization. D.B. and B.F. designed and performed experiments on HL60 cells. B.F. and K.A. designed and performed experiments on peripheral blood cells. Sa.G. provided calibration beads and characterized the calibration reference. B.F. developed the algorithm for rheological cell characterization. B.F. and St.G. established and carried out the statistical data analysis. O.O. and K.A. supervised the project. B.F. and O.O. wrote the manuscript. All authors reviewed the manuscript.

## Additional information

**Competing interests:** B.F., F.C., K.A., D.B., Sa.G and St.G. declare no competing interest. O.O. is shareholder of Zellmechanik Dresden GmbH distributing real-time deformability cytometry; Zellmechanik Dresden GmbH owns a patent for Real-Time Deformability Cytometry (RT-DC): EU patent under the number EP 30 036 520 B1.

