## [Peer Review File · Nature Communications]

Reviewers' comments:

Reviewer #1 (Remarks to the Author):

The authors present experimental results on the deformation of individual cells passing through a section of a microchannel. They propose that such mechanical phenotyping can be useful as a diagnostic of basic cell properties. The authors use image processing of the even and odd modes of the shape (viewed in its midplane) and comparison with previous theory they have reported in order to extract an "effective" elastic modulus and "effective" viscosity of the cell. In general, the paper is well written and the figures contain a lot of data. There were a lot of panels in each figure, so that some of the fonts were unfortunately small, but with enough squinting I was more or less able to follow the main results. The model is never really explained and I really think the paper needs a few sentences and an equation or two stating clear what is the mathematical model used to deduce the "effective properties". I tried to look up a few of the recent papers from the senior author has published ideas leading up to this paper but I was not able to understand what kind of uncertainty is associated with extracting the effective properties (and quite a few of the figure do not seem to indicate how many cells were measured for each data point).

I think the paper is interesting, and likely publishable in Nature Communication. But it really needs a revision so a reader can better judge how well the results compare with other independent measurements of the mechanical properties. A revision should also clarify the questions below which left me a little baffled in places and wondering exactly what to trust about the approach and what items might be significantly model dependent (maybe none, maybe all but since we are told little about the model how is one to judge).

As I read and thought about the paper I could not help but wonder:

- 1) I guess the authors are assuming that each deformed cell is nicely centered in the channel? Presumably this is not necessarily the case so how large an error might this make?
- 2) Without being told anything specific about the Fourier representation of the shape it is impossible to really understand what it means on p. 4 that the coefficients b_k have been dropped, which I suppose could be stated explicitly that they are set to zero. But why? Without knowing what are the b_k what is a reader to do?
- 3) Bottom of p. 4: I do not at all follow the argument that the d_{odd} determine the strain function? Why should this be so? The previous paragraph ties the even coefficients to cell deformation and presumably there is strain there? This seems important since it is an argument establishing how the stress-strain relationship is established. I realize the authors have likely thought a lot about this but this point is unclear and do they really mean the actual strain of the cell membrane is only determined by the odd coefficients? Almost on mathematical grounds I should think this has to be false.
- 4) Page 6: Young's modulus is given to 3 significant figures while the viscosity is given to 4 significant figures. Somehow I think this is likely wrong. If they don't know precisely where in the cross section is the cell then does not this introduce a significant uncertainty?
- 5) What exactly is the "effective modulus"? How does it compare with other measurement of the modulus in the literature? The same questions apply to the viscosity? As far as I can tell we are never told the answers to these questions. This comparison has to be made clearly so a reader can obtain some understanding of the utility of the method and possible errors that it might be hiding. Of course, the authors could try experiments with gel particles whose modulus and elasticity could be independently measured but apparently this was not done. It would be best if it was so that the methodology was properly calibrated.
- 6) The reader should really be told in some form in the main text the mathematical model from which the material properties are determined since this is central to understanding the obtained properties.
- 7) As far as I can tell the time scales τ are only defined in the figure captions but to read the main text they should be clearly defined there. Maybe I missed it.

Reviewer #2 (Remarks to the Author):

In this study, deformations of cells inside (and outside) a microfluidic channel are continuously (dynamically) monitored, and from the time course of the cell deformation, deformation parameters and cell mechanical parameters are extracted.

The authors find that cells elongate at the channel entrance, and round up and assume a more bullet-like shape as they travel along inside the channel. By performing a Fourier analysis of the cell shape, the time course of the cell rounding can be numerically represented by the even shape modes, and the time course of the bullet-like deformation can be numerically represented by the odd shape modes. The authors find that inside the channel, the cell deformations from the even modes decrease exponentially, and the deformations from the odd shape modes increase exponentially. The authors argue that this is due to a "slow increase in hydrodynamic shear stress outside the channel towards a maximum at the inlet and a sudden but smaller step stress inside the constriction". It is unclear what the differences between hydrodynamic shear stress and step stress might be. The authors then assume that the cell deformation (expressed as a circularity parameter) at the end of the channel for a given shear stress corresponds to an apparent Young's modulus of the cell. Under the further assumptions of linear material properties and a step-like increase of the stress inside the channel, the authors extract an apparent cell viscosity from the exponential time constant of the cell deformations and the Young's modulus. The remainder of the study uses the shape parameters and derived cell mechanical parameters to analyze a cell line under different treatment conditions, and whole blood cells.

What is missing from the study is an understanding of the cell shape and its dynamics in response to the hydrodynamic stress versus time profile outside and inside the microfluidic channel. In particular, it is unclear why cells first elongate and then deform into a bullet-like shape. The hydrodynamic shear stress profile in the flow direction seen in Fig. 1 may perhaps correspond somehow to the cell elongation over time (even shape modes) but not the bullet-like shape dynamics (odd modes). Shear stresses are obtained from FEM computations without considering the presence of a cell (and its particular shape) inside a channel, which is a questionable simplification. Moreover, the cell deformation as computed from the circularity index (in fig. 1) is approximately the same at the channel entrance and near the end of the channel despite different shear stress values. This calls into question a previously published method for computing the cell's apparent Young's modulus, on which this study rests for computing cell viscosities. Finally but importantly, the study lacks control and validation experiments, e.g. measurements at different flow speeds to test for linearity, or measurements with elastic beads or oil droplets.

Reviewer #3 (Remarks to the Author):

The authors report the development of time-dependent measurements using the real-time deformability cytometry platform. By using Fourier analysis of the deformed shape of a cell as it transits in a channel with a viscous fluid they were able to extract two separate time-dependent deformation metrics, which they tie to physical parameters of cells using a fluid stress model. The authors are able to measure the viscoelastic properties of red blood cells, granulocytes, and lymphocytes in a single experiment which allows improved comparisons. Interestingly, at the population level differences in the mean elasticity and viscosity of B-lymphocytes and CD4+ T-lymphocytes was observed. Overall, this is a solid work, however, the final claim that population level differences in these lymphocytes could lead to accurate sorting, which requires classification at the single-cell level is not supported by the data.

Major comments:

1) In the final paragraph of the discussion the authors write: "In this context, we provide evidence that dRT-DC is capable to discriminate between isolated leukocytes of different lineages including

B- and T-lymphocytes with statistical significance.” The authors then go on to discuss how this may be important to sorting. The authors should be more careful with these claims. Accurate sorting requires classification at the single-cell level, however, the authors only report that at the population level the means of the two populations are slightly different. However, no histograms of the populations are provided or some analysis of classification accuracy (ROC, AUC of the ROC, classification accuracy etc.). In order to make claims of distinguishing between single-cells between these populations this additional data should be provided. Alternatively the authors should only claim what is supported by the data, that the means of the population are statistically different, which does not apply to classification accuracy.

2) In a related matter, in Fig. 4a-d standard error of the mean is used. How many cells are used in these calculations? How much do the distributions from different cell types overlap? As discussed, it can be misleading for applications in classification of individual cells to use the SEM of the population, as population distributions can overlap significantly but still have a different mean and tight SEM if n is large enough.

3) It also, looks like based on the methods section that only CD4+ T-lymphocytes were measured (not CD8+ T-lymphocytes). This should be clarified.

4) An important reference for measuring viscoelastic properties in DC geometries is missing: Kumar et al. Biophysical Journal 111, 2039–2050, November 1, 2016. Many similar experiments are performed (e.g. cytochalasin D treatment), and comparisons should be made with this work.

5) How is stress defined as a function of distance in the channel along a certain streamline. This is not very accurate as the stress a cell experiences will depend on location in the channel and dimensions of the cell, especially since the cell approaches the dimensions of the channel. When the cell is displacing the fluid as it flows through the channel the stress on the cell surface may be much higher than shown. It seems simulations are only performed for a single cell diameter – this should be clarified?

6) In a related matter, the different velocities in Fig.4 suggest different paths / different stress distributions. How is this taken into account with calculating Young’s modulus and subsequent viscosity parameters which depend on Young’s modulus?

Other comments:

Figure 2c and Figure 2d are out of order. This might be intended to be read in a clockwise fashion, in addition to keeping the widths of a-d and b-c the same, but it is confusing.

Figure S3 should be broken into two different plots. Two y-axes are used, but the points don’t really occupy the same x-axis. Colocation of the series on the same plot invites comparison between the two series, but with only two points in a series, they can be made to look as similar or dissimilar as desired.

Given how the overlapping traces are shown in Figure 2b, it is difficult to understand aspects of the single-cell behavior. For example, for a given axial position, are the single-cells normally distributed? Do the traces cross over each other? The traces seem to move together with a given sigma about the mean, but the data would be more interesting without much trace crossover.

Reviewers' comments:

Reviewer #1 (Remarks to the Author):

Reviewer:

The authors present experimental results on the deformation of individual cells passing through a section of a microchannel. They propose that such mechanical phenotyping can be useful as a diagnostic of basic cell properties. The authors use image processing of the even and odd modes of the shape (viewed in its midplane) and comparison with previous theory they have reported in order to extract an “effective” elastic modulus and “effective” viscosity of the cell. In general, the paper is well written and the figures contain a lot of data. There were a lot of panels in each figure, so that some of the fonts were unfortunately small, but with enough squinting I was more or less able to follow the main results. The model is never really explained and I really think the paper needs a few sentences and an equation or two stating clear what is the mathematical model used to deduce the “effective properties”. I tried to look up a few of the recent papers from the senior author has published ideas leading up to this paper but I was not able to understand what kind of uncertainty is associated with extracting the effective properties (and quite a few of the figure do not seem to indicate how many cells were measured for each data point).

Author response:

We thank the reviewer for the time reviewing our manuscript and the overall positive feedback. We fully agree with the statement that the model for extracting material properties from our data is not very well explained and improved this part in the revised version as highlighted in the point-by-point response.

Reviewer:

I think the paper is interesting, and likely publishable in Nature Communication. But it really needs a revision so a reader can better judge how well the results compare with other independent measurements of the mechanical properties. A revision should also clarify the questions below which left me a little baffled in places and wondering exactly what to trust about the approach and what items might be significantly model dependent (maybe none, maybe all but since we are told little about the model how is one to judge).

As I read and thought about the paper I could not help but wonder:

1) I guess the authors are assuming that each deformed cell is nicely centered in the channel? Presumably this is not necessarily the case so how large an error might this make?

Author response:

We gratefully acknowledge this question from the reviewer. In general, the cells are very well centred in the channel since we use hydrodynamic focussing for alignment of the cells in the direction of flow. As we don't address this in the microfluidic layout of Figure 1 of the main manuscript, we verified the cell alignment by analysing the lateral distribution of cells at the outlet of the channel.

For the data from Figure 3 this leads to a normal distribution of cells perpendicular to the channel axis as shown in the Q-Q plots of Supplementary Fig. 10 for a) wildtype HL60 cells, b) HL60 cells after treatment with 0.25% (v/v) DMSO and c) HL60 cells after treatment with 1 μ M CytoD (top row). Analysing the standard deviation of this distribution (with respect to the channel centre axis) we find $\sigma=0.68 \mu\text{m}$ (wildtype), $\sigma=0.57 \mu\text{m}$ (DMSO control) and $\sigma=0.95 \mu\text{m}$ (CytoD) verifying the efficiency of the hydrodynamic focussing and confirming a narrow distribution around the center line.

In addition, we analysed the apparent elastic Young's modulus and the apparent viscosity as a function of the lateral channel position. Here the standard deviation of the elastic Young's modulus is found for (a) and (b) to $\sigma=0.30 \text{ kPa}$ (wildtype and DMSO control) and (c) to $\sigma=0.26 \text{ kPa}$ (CytoD) while for the viscosity we observe (a) $\sigma=2.80 \text{ Pa}\cdot\text{s}$ (wildtype), (b) $\sigma=2.73 \text{ Pa}\cdot\text{s}$ (DMSO control) and (c) $\sigma=2.99 \text{ Pa}\cdot\text{s}$ (CytoD). Performing a signed-rank test we find no correlation between the lateral channel position and the apparent elastic Young's modulus as well as the apparent viscosity (correlation coefficient $\rho<0.09$). This confirms that there is no systematic offset for small displacements from the channel centre axis.

For addressing the question of position dependency we have added Supplementary Fig. 10 in the Supplementary Material and included the following paragraph in the Methods section (Analytical model) on page 12:

"Application of this simple model to calculate the Young's modulus requires an alignment of the cells in the center of the channel³¹. We verified this condition by analyzing the apparent elastic Young's modulus and the apparent viscosity as a function of the cell displacement from the channel center and find a standard deviation $\sigma<1 \mu\text{m}$ for all conditions. Using a Spearman rank correlation test we find correlation coefficients of $\rho<0.09$ for all conditions which confirms that there is no correlation between the rheological parameters and lateral cell position (Supplementary Fig. 10)."

Reviewer:

2) Without being told anything specific about the Fourier representation of the shape it is impossible to really understand what it means on p. 4 that the coefficients b_k have been dropped, which I suppose could be stated explicitly that they are set to zero. But why? Without knowing what are the b_k what is a reader to do?

Author response:

Thanks a lot for this very helpful comment. The reviewer is right when saying that we have not stated our procedure of dealing with the b_k in full extend. The representation in Figure 2 of the original manuscript has been calculated for setting b_k equal to zero. Here, we utilized the fact that the b_k represent the angular orientation of a cell inside the channel as indicated in Supplementary Fig. 3a. Since a sheath flow focusses the cells in the centre of the channel, the cells align along the symmetry axis with respect to the direction of flow (Figure 1a). For the mean cell shape this is shown in Figure 2c and 2d of the original manuscript. Assuming a narrow distribution of cells with respect to the channel center line the contribution of b_k to the reconstruction of the overall cell shape is close to zero (Supplementary Fig. 3c).

Supplementary Fig. 3c also indicates that the simplification of setting $b_k=0$ is only valid for cells that are centered exactly in the middle of the microfluidic channel. Any deviation from this alignment will introduce an asymmetry rendering the b_k important for shape reconstruction (the further away from the central channel axis the higher the impact of b_k coefficients).

Since we aim to perform single cell rheology, small deviations from the channel centre line might be observed especially in complex samples as whole blood. This leads to slightly asymmetric cell deformation and a significant contribution of b_k towards the construction of cell shape.

For consistency the b_k are now used for all shape reconstructions in Figures 2, 3 and 4 and the following sentence on page 4 has been dropped:

“Of note, Fourier components b_k have safely been dropped in the reconstruction as they represent the angular cell orientation which is close to zero due to the orientation of the cells along the axial position z inside the channel (Supplementary Fig. S2c).”

and is replaced by:

“Of note, Fourier coefficients b_k represent the angular orientation of a cell inside the channel revealing a mean value close to zero and are of importance for single cells which are slightly displaced from the channel center (Supplementary Fig. 3c).”

Reviewer:

3) Bottom of p. 4: I do not at all follow the argument that the d_{odd} determine the strain function? Why should this be so? The previous paragraph ties the even coefficients to cell deformation and presumably there is strain there? This seems important since it is an argument establishing how the stress-strain relationship is established. I realize the authors have likely thought a lot about this but this point is

unclear and do they really mean the actual strain of the cell membrane is only determined by the odd coefficients? Almost on mathematical grounds I should think this has to be false.

Author response:

The reviewer is completely right when saying that both, d_{even} and d_{odd} represent a strain function.

For our line of thoughts it has to be taken into account that the stress distribution on the cell surface inside a microfluidic channel is described by Legendre polynomials¹. This stress distribution leads to a symmetric Lagrangian strain tensor where surface displacements are a function of the radius and the polar angle (Lurie, A. I. 2005. Theory of Elasticity. Springer, New York. 36).

Figure R1: Description of cell strain. a) Relative displacement u_i/r_0 for radial component u_r (solid line) and polar component u_θ (dashed line) derived for an elastic sphere (red) and shell (black). b) Deformation vs. area plot showing agarose bead measurements (blue) and HL-60 cells (red). Dashed lines indicate magnitude of maximal eigenvalue of strain tensor averaged over sphere surface (adapted from ref.¹).

The dependency between polar angle and relative surface displacement is indicated in Fig. R1a where calculations have been done for an elastic sphere and elastic shell using an analytical model published earlier¹. Calculating the magnitude of the maximal eigenvalue of the strain tensor averaged over the sphere's surface allows to link strain into the deformation vs. cell area phase space as shown in Fig. R1b.

It has to be emphasized that we base the calculation of cell deformation on the parameter circularity:

$$d = 1 - c = 1 - \frac{2\sqrt{\pi A}}{p}, \#(1)$$

which is more representative for bullet-like shape changes than a conventional strain parameter, e.g. $\varepsilon = \Delta/l$. In fact, it is straight forward to construct a transition from an initially circular shape into a bullet shape with $\varepsilon = 0$ but $d > 0$.

Fig. R1b also shows that the relationship between the magnitude of the maximal eigenvalue of the strain tensor and cell deformation is not-trivial. This important information has not been included in the initial form of the manuscript.

We would like to emphasize that our analytical model presented in Mietke *et al.* allows to extract an elastic Young's modulus from circularity under the assumption of steady-state deformation and an initially spherical cell shape¹. For this reason, we were not able to analyse time-dependent material properties and also not to

characterize cells deviating from a sphere in bulk solution. In addition, the circularity is not meaningful in the inlet region of the channel since the large hydrodynamic stresses require the full Lagrangian strain tensor incorporating the changes in the fluid flow around the deformed object.

Our current manuscript provides now a possibility to decouple cell deformation in response to the inlet peak stress from cell deformations originating from the constant stress inside the channel using Fourier decomposition of cellular shape modes.

The superposition of these two dynamic processes allows us to focus on the channel deformation only and to consider d_{channel} as the response of the cell to a constant stress as required in creep compliance experiments. All inlet effects can safely be ignored. The characteristic time τ_{channel} to reach the steady-state is obtained from an exponential fit (Equation 5) used to calculate the viscosity of the cell.

For clarification we have replaced the following paragraph on page 4:

“In summary, shape mode analysis allows to disentangle the response of the cells to an inlet peak stress and a constant hydrodynamic stress inside the channel (Fig. 1c). This implies that d_{odd} can be seen as the strain function ε of a cell in response to a step stress σ , where σ is given by the hydrodynamic shear stress inside the constriction³¹. Having established the stress-strain relationship, rheological parameters can in principle be extracted from the data.”

by:

“In summary, shape mode analysis allows to disentangle the cell response to the inlet peak stress σ_{inlet} from the channel stress σ_{channel} (Fig. 1c). This implies that d_{even} can be seen as the deformation trace d_{inlet} representing shape dynamics caused by σ_{inlet} while d_{odd} can be understood as the deformation trace d_{channel} due to the constant stress σ_{channel} (Fig. 2c). The latter can be approximated by a step-function and is given by the hydrodynamic stress around the cell inside the constriction (Supplementary Fig. 4). Applying an analytical model published earlier d_{channel} enables to establish a stress-strain relationship³¹. Therewith, rheological parameters can directly be determined from the channel data representing a creep-compliance experiment³⁶.”

Reviewer:

4) Page 6: Young's modulus is given to 3 significant figures while the viscosity is given to 4 significant figures. Somehow I think this is likely wrong. If they don't know precisely where in the cross section is the cell then does not this introduce a significant uncertainty?

Author response:

We thank the reviewer for this comment. We corrected it and give all our results with three significant figures. As discussed in response to the first question the position of the cell with respect to the channel axis has no systematic effect on the elastic Young's modulus or viscosity. The uncertainty stated in the text originates from the comparison between two conditions, respectively, which have been analysed using linear mixed models. Therefore this uncertainty represents the standard error of the mean of the three replicates for each of our experimental analysis.

Reviewer:

5) What exactly is the “effective modulus”? How does it compare with other measurement of the modulus in the literature? The same questions apply to the viscosity? As far as I can tell we are never told the answers to these questions. This comparison has to be made clearly so a reader can obtain some understanding of the utility of the method and possible errors that it might be hiding. Of course, the authors could try experiments with gel particles whose modulus and elasticity could be independently measured but apparently this was not done. It would be best if it was so that the methodology was properly calibrated.

Author response:

We thank the reviewer for this question. We use the apparent modulus and viscosity since the actual values always depend on the measurement system and the loading rate applied. This fact was emphasized in a recent publication where different methods for cell mechanical properties have been compared³. We also address this issue in the discussion where we compare previous results on HL60 cells with dRT-DC data. Here, we find a lower elastic Young's modulus (0.4 kPa versus 1.5 kPa) which we can potentially attribute to strain-stiffening.

In the light of the excellent suggestion to use gel particles for calibration we performed dynamic RT-DC measurement with poly-acrylamide microgel beads of a fixed elastic modulus following ref.⁵. In this recent publication the authors demonstrate the generation of monodisperse beads with adjustable elasticity. For calibration beads with a polymer content of 7.9% (w/v) and a corresponding elastic modulus of 1.5 ± 0.5 kPa were obtained from Salvatore Girardo the main author of the publication.

Having a mean diameter of 15.2 ± 0.8 μm dRT-DC on these beads was performed in $20 \mu\text{m} \times 20 \mu\text{m}$ and $30 \mu\text{m} \times 30 \mu\text{m}$ channels at a flow rate of 16 nl/s and 128 nl/s, respectively. Following the protocol provided in the publication⁵ a deformation of approximately $d=0.02$ was adjusted before we applied our analytical model to extract the elastic modulus¹. For both, $20 \mu\text{m} \times 20 \mu\text{m}$ and $30 \mu\text{m} \times 30 \mu\text{m}$ channels, we obtain an elastic modulus of 2.2 ± 0.03 kPa and 2 ± 0.04 kPa, respectively (Supplementary Fig. 11).

These values are slightly higher than the reported results of 1.5 ± 0.5 kPa but within or close to the standard error of the mean. Two possible reason can be attributed to that. First of all the bead production reveals a slide batch-batch variation as shown in Table S4 of ref⁵. Second, dRT-DC specifically incorporates the initial shape of the beads leading to a reduced deformation magnitude compared to RT-DC and by that to a slightly elevated and more correct apparent elastic Young's modulus.

The authors from Girardo *et al.* have also been measuring the relaxation time of the ensemble average over many beads inside the channel and obtained a value of 0.12 ± 0.02 ms⁵. Remarkably, this is close to the value of 0.36 ± 0.01 ms ($20 \mu\text{m} \times 20 \mu\text{m}$ channel) and 0.12 ± 0.01 ms ($30 \mu\text{m} \times 30 \mu\text{m}$ channel) for single bead rheology in dRT-DC. In summary, these results confirm the possibility of calibrating dRT-DC and its utilization to extract rheological parameters from single cell measurements.

Based on this calibration we realized for some conditions that our analysis software (ShapeOut) extracts the wrong channel diameter from the measurement configuration files. This resulted in an overestimation of the apparent Young's modulus and subsequently the apparent viscosity. This made us re-analysing the

data shown in Figure 3 and Figure 4 of the main manuscript. While deformation values and characteristic times are not affected, applying a channel diameter which is off by a factor of up to 1.5 (20 μm x 20 μm instead of 30 μm x 30 μm) resulted in an error in the magnitude of the elastic Young's modulus and viscosity. We would like to emphasize that this conversion error had only an impact on the absolute amplitude of the rheological parameters but not on the observed statistical trends. Since statistical analysis is based on linear mixed models comparing replicates of one group of samples to another group these relative differences are not affected.

We thank the reviewer for this great suggestion of using beads for calibration, we corrected our error and have been addressing the calibration of dRT-DC by adding the following paragraph on page 12 of the Methods section:

“Calibration of dynamic RT-DC

Calibration of dRT-DC has been carried out using standardized elastic poly-acrylamide microgel beads from ref. ⁴⁸. The rheology of beads with a mean diameter of 15.2 ± 0.8 μm and an elastic modulus of 1.5 ± 0.5 kPa has been characterized in 20 μm x 20 μm and 30 μm x 30 μm channels at a flow rate of

16 nl s^{-1} and 128 nl s^{-1} , respectively. Using the peak channel deformation \hat{d}_{channel} and applying our analytical model we obtain an apparent elastic Young's modulus of 2.20 ± 0.03 kPa and 2.00 ± 0.04 kPa (Supplementary Fig. 11). The slight deviation from the calibration standard is still within or close to the standard error of the mean and could be attributed to batch-to-batch variation as discussed in ref ⁴⁸. Interestingly, the authors from Girardo *et al.* also performed measurements on the relaxation time using ensemble averages over hundreds of beads. They find a value of 0.12 ± 0.02 ms, which is remarkably close to 0.36 ± 0.01 ms (20 μm x 20 μm channel) and 0.12 ± 0.01 ms (30 μm x 30 μm channel) for single beads.”

Reviewer:

6) The reader should really be told in some form in the main text the mathematical model from which the material properties are determined since this is central to understanding the obtained properties.

Author response:

We gratefully acknowledge the comment from the reviewer. We have included a detailed explanation of the mathematical model in the Methods section (Analytical model) of the revised manuscript. In addition, we have changed the sequence of the sub-sections. Now we first explain the hydrodynamic simulations which are being used to calculate the hydrodynamic stress on the cell surface in steady-state. This is illustrated in Figure R2, which we have also incorporated in the revised Supplementary Material (Supplementary Fig. 4).

Figure R2: Finite element method simulation of steady-state surface stress. The graph shows the shear stress distribution on a sphere of a diameter of $15.9 \mu\text{m}$ flowing through a channel of $30 \mu\text{m} \times 30 \mu\text{m}$ cross-section. Calculations have been conducted under steady-state conditions for a flow rate of 8 nl/s and the average shear stress on the sphere surface has been calculated to 142 Pa and a shear rate of 5.100 1/s was obtained. Finite element method simulations have been carried out using COMSOL.

In the main manuscript, we have replaced the following paragraph on page 5:

“Next, our findings on the systems symmetry are used to extract rheological parameters from our data. For traces of $\Delta d_{\text{channel}}$ revealing a steady state deformation (Fig. 3a, bottom), an analytical model can be used to calculate an apparent elastic Young’s modulus E ^{1,6}.”

by:

“Next, we use the channel deformation trace to extract rheological parameters from our data. For traces of d_{channel} revealing a steady-state (Fig. 3a, bottom), an analytical model can be used to obtain an apparent elastic Young’s modulus E ^{31,32} and an apparent viscosity η assuming linear viscoelasticity (see Methods) ²⁰. Briefly, finite element method (FEM) simulations are carried out to obtain the hydrodynamic surface stress distribution of a cell moving in a microfluidic channel (Supplementary Fig. 4). For a cell moving within a $30 \mu\text{m} \times 30 \mu\text{m}$ channel at a flow rate of 8 nl/s a mean surface shear stress of 142 Pa (dashed line in Fig. 3a, bottom) is determined. Using this stress distribution the cell deformation can be predicted and mapped to an apparent Young’s modulus ³¹.”

In the Methods section (Analytical model) on page 12 we have replaced:

“The relaxation time τ_{channel} is considered the creep-compliance response to the constant stress σ that allows to analyze the data using a Kelvin-Voigt model with $\tau_{\text{channel}} = \eta/E$, where η is the apparent viscosity.”

by:

“For calculation of cell rheological properties, we use the channel deformation trace d_{channel} only. Here, cells reach a steady-state deformation \hat{d}_{channel} , which is defined by the balance of normal and shear forces inside the constriction. In this steady-state we derive the cellular surface stress using FEM simulations as described above. The apparent elastic Young’s modulus E is then

calculated using an analytical model published earlier¹. Briefly, by coupling the hydrodynamic stress distribution in the channel to linear elasticity theory the deformation of a suspended spherical object can be predicted and mapped to the experimental conditions. Calculations are being performed for the mean size of each cell type presented in this study including flow-rate dependency and accompanying shear-thinning behavior of the surrounding medium. While d_{channel} does not represent a simple strain function, e.g. $\varepsilon \approx \Delta/l$, due to the parabolic hydrodynamic stress distribution, it is sufficient to represent the dynamic response of the cell to the step stress inside the channel and to derive a relaxation time from the fit of the channel deformation trace d_{channel} to Equation 5.

This relaxation time τ_{channel} can be considered the characteristic time for the creep-compliance response of the cell to a constant stress σ_{channel} (dashed line in Fig. 3a, bottom) inside the channel. Here, an apparent viscosity η can be calculated using a simple Kelvin-Voigt model with:

$$\tau_{\text{channel}} = \frac{\eta}{E}, \quad (7)$$

where E is the apparent elastic Young's modulus."

Reviewer:

7) As far as I can tell the time scales tau are only defined in the figure captions but to read the main text they should be clearly defined there. Maybe I missed it.

Author response:

We thank the reviewer for the comment. The characteristic times had already been defined on page 5, but could maybe not clearly be identified due to the unprecise discussion about stress and strain. By unambiguously introducing inlet deformation d_{inlet} and channel deformation d_{channel} including the steady-state deformations \hat{d}_{inlet} and \hat{d}_{channel} the relationship between strain, stress and corresponding time scale τ should be more obvious now.

We have been addressing this comment by replacing the following part on page 4:

"The inlet deformation trace is described by the peak deformation at the inlet relative to the steady-state in the channel and the corresponding relaxation time τ_{inlet} (Fig. 3a, top). The response of the cell to the constant shear stress inside the constriction reconstructed from the first five odd shape modes including a_0 is given by the steady-state deformation at the channel outlet relative to the inlet and the characteristic time τ_{channel} (Fig. 3a, bottom). These timescales originate from an exponential fit to our reconstructed data traces assuming a linear cell response (see Methods)."

by:

"For this inlet deformation trace, the local maximum defines the peak deformation \hat{d}_{inlet} while the relaxation into a steady-state is quantified by the corresponding time constant τ_{inlet} (Fig. 3a, top). The response of the cell d_{channel} to the constant shear stress σ_{channel} inside the constriction is reconstructed from a_0 and the first five odd shape modes. Reaching a maximum at the channel outlet, d_{channel} is described by the steady-state deformation \hat{d}_{channel} relative to the inlet and the characteristic time constant τ_{channel} (Fig. 3a, bottom). Both timescales, τ_{inlet} and τ_{channel} , originate from an exponential fit to our reconstructed data traces (yellow lines in Fig. 3a) assuming a linear cell response (see Methods)."

Reviewer #2 (Remarks to the Author):

In this study, deformations of cells inside (and outside) a microfluidic channel are continuously (dynamically) monitored, and from the time course of the cell deformation, deformation parameters and cell mechanical parameters are extracted. The authors find that cells elongate at the channel entrance, and round up and assume a more bullet-like shape as they travel along inside the channel. By performing a Fourier analysis of the cell shape, the time course of the cell rounding can be numerically represented by the even shape modes, and the time course of the bullet-like deformation can be numerically represented by the odd shape modes. The authors find that inside the channel, the cell deformations from the even modes decrease exponentially, and the deformations from the odd shape modes increase exponentially. The authors argue that this is due to a “slow increase in hydrodynamic shear stress outside the channel towards a maximum at the inlet and a sudden but smaller step stress inside the constriction”. It is unclear what the differences between hydrodynamic shear stress and step stress might be. The authors then assume that the cell deformation (expressed as a circularity parameter) at the end of the channel for a give shear stress corresponds to an apparent Young’s modulus of the cell. Under the further assumptions of linear material properties and a step-like increase of the stress inside the channel, the authors extract an apparent cell viscosity from the exponential time constant of the cell deformations and the Young’s modulus. The remainder of the study uses the shape parameters and derived cell mechanical parameters to analyze a cell line under different treatment conditions, and whole blood cells.

Author response:

We appreciate the positive summary of our work and thank the reviewer for the suggestions and comments. The reviewer asks an important question when pointing towards the difference between the hydrodynamic shear stress and step stress. The cell inside the channel experiences a hydrodynamic stress which has a normal component in flow direction and a tangential shear stress component. Under steady-state conditions inside the channel the net force on the cell is equal to zero and the integrated normal stress components equal the shear stress components¹. Since both components are of the same order of magnitude we were only referring to a hydrodynamic shear stress inside the channel.

This has been misleading and discrimination between hydrodynamic normal and shear stress has not been very well explained in the original version of the manuscript. We improved on this by mainly talking about hydrodynamic stress. Only when specifically addressing shear stress, e.g. on page 2 we use this term.

Secondly, the reviewer asks about the difference between step stress and shear stress. The step stress is the constant hydrodynamic stress inside the channel due to the Poiseuille flow profile. In the manuscript we show that a Fourier decomposition of cellular shapes and subsequent reconstruction from odd modes leads to a deformation trace d_{channel} which represents the cell response to the constant hydrodynamic stress (step stress) inside the channel.

For better understanding of this part we have replaced the paragraph on page 4:

“In summary, shape mode analysis allows to disentangle the response of the cells to an inlet peak stress and a constant hydrodynamic stress inside the channel (Fig. 1c). This implies that d_{odd} can be seen as the strain function ε of a cell in response to a step stress σ , where σ is given by the hydrodynamic shear stress inside the constriction¹. Having established the stress-strain relationship, rheological parameters can in principle be extracted from the data.”

by:

“In summary, shape mode analysis allows to disentangle the cell response to the inlet peak stress σ_{inlet} from the channel stress σ_{channel} (Fig. 1c). This implies that d_{even} can be seen as the deformation trace d_{inlet} representing shape dynamics caused by σ_{inlet} while d_{odd} can be understood as the deformation trace d_{channel} due to the constant stress σ_{channel} (Fig. 2c). The latter can be approximated by a step-function and is given by the hydrodynamic stress around the cell inside the constriction (Supplementary Fig. 4). Applying an analytical model published earlier d_{channel} enables to establish a stress-strain relationship³¹. Therewith, rheological parameters can directly be determined from the channel data representing a creep-compliance experiment³⁶.”

Reviewer:

What is missing from the study is an understanding of the cell shape and its dynamics in response to the hydrodynamic stress versus time profile outside and inside the microfluidic channel. In particular, it is unclear why cells first elongate and then deform into a bullet-like shape. The hydrodynamic shear stress profile in the flow direction seen in Fig. 1 may perhaps correspond somehow to the cell elongation over time (even shape modes) but not the bullet-like shape dynamics (odd modes).

Author response:

We gratefully acknowledge the comment from the reviewer and agree that the physical origin of cell elongation at channel inlet and bullet shape inside the channel was not well explained.

Performing finite element method simulations and extracting the hydrodynamic velocity component v_z in flow direction shows an increase in velocity magnitude until the channel inlet at $z=0 \mu\text{m}$. The increase in magnitude can be understood from the reduction in cross-section from the channel reservoir to the constriction leading to a higher velocity. Inside the channel between $z=0 \mu\text{m}$ and $z=300 \mu\text{m}$ the velocity is constant (Fig. R3a). This is expected due to the Poiseuille flow profile inside the channel. Beyond the end of the channel at $z=300 \mu\text{m}$ the cross-section increases and the velocity is reduced.

Figure R3: Velocity profile and hydrodynamic shear stress along channel center line.

a) Velocity magnitude (all vector components) for three typical flow rates of 4 nl/s, 8 nl/s and 16 nl/s reaching a plateau inside 30 μm x 30 μm channel. The channel starts at $z = 0$ μm and ends at $z = 300$ μm. b) Velocity gradient in direction of flow. c) Hydrodynamic shear stress showing a peak at the channel inlet ($z = 0$ μm) and channel outlet ($z = 300$ μm) and a constant plateau inside the constriction. Calculations have been done without a cell using finite element method simulations in COMSOL.

When taking the derivative of the velocity in flow direction dv/dz we find a peak at the channel inlet ($z=0$ μm) and the channel outlet ($z=300$ μm, Fig. R3b). This acceleration in z -direction leads to an elongation of the cell in the direction of flow, which is also reflected in the hydrodynamic shear stress peak at the inlet ($z=0$ μm) and outlet ($z=300$ μm, Fig. R3c)

Inside the constriction and sufficiently away from the inlet the hydrodynamic environment is different. Here, the cells experience a velocity gradient v_y perpendicular to the direction of flow. This gradient results from the no-slip boundary condition and the parabolic flow profile with the highest velocity in the centre and leads to a constant stress inside the constriction (between $z=0$ μm and $z=300$ μm). For this representation finite element method simulations have been done without the presence of the cell (Fig. R3c). The assumption of a constant channel stress is still valid as the disturbance of the flow profile by the cell can be neglected (Diploma Thesis Alexander Mietke).

The cell reaches a steady-state when normal forces in flow direction balance tangential shear forces and the cell responds with the bullet shape ¹. Please note that although the channel has a square cross-section, the assumption of a parabolic flow is justified since the profile is only perturbed in the corners and as the circular cross-section can be mapped to the square cross-section using the equivalent channel radius.

For a more detailed explanation of the cell elongation at the channel inlet and the bullet like shape in the channel we included Figure R3 in the Supplementary Material (Supplementary Fig. 2) and also improved the explanation on page 3 of the main part where we replaced:

" The velocity gradient in flow direction leads to a peak stress σ_{inlet} directly at the inlet and outlet while a smaller stress amplitude σ_{channel} is found inside the channel."

by:

"The velocity gradient in flow direction at the inlet and outlet (Supplementary Fig. 2a and b) leads to a peak stress σ_{inlet} while a smaller stress amplitude σ_{channel} is found inside the channel (Fig. 1c, bottom and Supplementary Fig. 2c)."

and on page 4 we replaced:

“This behavior originates from the fluid velocity gradient in flow direction with the maximum in shear stress at the inlet (Fig. 1c). The corresponding cell contour possesses the same symmetry of shape changes along the channel axis as the even shape modes (a_0), a_2 ... a_8 (Fig. 2a and left panel in Fig. 2c).”

by:

“This behavior originates from the fluid velocity gradient in flow direction (Supplementary Fig. 2b) yielding a maximum in hydrodynamic stress at the inlet (Fig. 1c and Supplementary Fig. 2c). The corresponding cell contour of an elongated cell possesses the same symmetry of shape changes along the channel axis as the even shape modes (a_0), a_2 ... a_8 (Fig. 2a and left panel in Fig. 2c).”

Reviewer:

Shear stresses are obtained from FEM computations without considering the presence of a cell (and its particular shape) inside a channel, which is a questionable simplification. Moreover, the cell deformation as computed from the circularity index (in fig. 1) is approximately the same at the channel entrance and near the end of the channel despite different shear stress values. This calls into question a previously published method for computing the cell's apparent Young's modulus, on which this study rests for computing cell viscosities.

Author response:

The reviewer is correct when assuming that the hydrodynamic stress in Fig. 1 is calculated without the presence of a cell. This was done to illustrate the hydrodynamic stress distribution along the symmetry axis of our microfluidic system. Here, the presence of the cell only locally disturbs the flow profile which has no impact on the laminar flow condition^{1,6}.

In contrast all rheological cell properties have been derived performing finite element method simulations in presence of a cell (Figure R2). This was not clearly outlined in the manuscript. In fact, calculations of rheological properties have been performed considering all mean cell sizes, the appropriate flow rates and viscosities of the medium (Table R1).

Channel cross-section (μm^2)	Flow rate (nl/s)	Cell diameter (μm)	Shear rate (1/s)	Hydrodynamic shear stress (Pa)	Dynamic viscosity (mPa s)
20 * 20	4	6.7	9904	221	25.0
20 * 20	8	6.7	18854	335	19.9
20 * 20	16	6.7	40857	551	15.1
20 * 20	4	10.0	9744	216	26.0
20 * 20	8	10.0	21510	363	19.3
20 * 20	16	10.0	39201	531	15.8
20 * 20	4	13.3	11538	234	27.8
20 * 20	8	13.3	23680	373	21.3
20 * 20	16	13.3	44667	617	17.9
30 * 30	4	10.0	2629	94	40.0
30 * 30	8	10.0	5259	147	31.3
30 * 30	16	10.0	10518	230	24.4
30 * 30	4	15.0	2621	93	42.0
30 * 30	8	15.0	5105	142	33.1
30 * 30	16	15.0	10243	223	25.9
30 * 30	4	20.0	2898	95	49.4
30 * 30	8	20.0	5864	149	38.2
30 * 30	16	20.0	11728	234	29.8

Table R1: Finite element method simulations of hydrodynamic flow profile around a cell. Calculations of shear rate, mean hydrodynamic shear stress and viscosity have been carried out for channel cross-sections of $20 \mu\text{m} \times 20 \mu\text{m}$ and $30 \mu\text{m} \times 30 \mu\text{m}$, varying flow rates and different cell diameters under steady-state conditions.

In addition, we also analysed the impact of cell shape on the flow profile and hydrodynamic stress distribution using finite element method simulations. Comparing a spherical cell and a bullet-like cell we find that the small strains (between 8% and 12%) in our system do not impact on the hydrodynamic stress distribution on the cell surface (Figure R4 and R5). In fact, the stress distributions are nearly the same. Calculations have been done for a sphere (Figure R4) and a bullet-like shape (Figure R5) of $7.5 \mu\text{m}$ radius each and in a channel with $30 \mu\text{m} \times 30 \mu\text{m}$ cross-section under steady-state conditions.

Figure R4: Hydrodynamic stress distribution around spherical cell in steady-state. The graph shows shear-stress distribution on a sphere with a mean radius of 7.5 μm (obtained from Fig. 2) in a 30 μm x 30 μm channel (left) and the corresponding velocity profile (right). Calculations have been done under steady-state conditions and the average shear stress was calculated to 83 Pa and a shear rate of 2.279 1/s was obtained. Finite element calculations have been carried out using COMSOL.

Figure R5: Hydrodynamic stress distribution around bullet-like cell in steady-state. The graph shows shear-stress distribution on a bullet-like cell with a corresponding radius of 7.5 μm (obtained from Fig. 2) in a 30 μm x 30 μm channel (left) and the corresponding velocity profile (right). Calculations have been done under steady-state conditions and the average shear stress was calculated to 83 Pa and a shear rate of 2.279 1/s was obtained. Finite element calculations have been carried out using COMSOL.

For improving this part of our manuscript we have include Table R1 in the Supplementary Material as Supplementary Table 1 and adapted the Methods section as follows (page 12):

“For calculation of cell rheological properties, we use the channel deformation trace d_{channel} only. Here, cells reach a steady-state deformation \hat{d}_{channel} , which is defined by the balance of normal and shear forces inside the constriction. In this steady-state we derive the cellular surface stress using FEM simulations as described above. The apparent elastic Young’s modulus E is then calculated using an analytical model published earlier¹. Briefly, by coupling the hydrodynamic stress distribution in the channel to linear elasticity theory the deformation of a suspended spherical object can be predicted and mapped to the experimental conditions. Calculations are being performed for the mean size of each cell type presented in this study including flow-rate dependency and accompanying shear-thinning behavior of the surrounding medium.”

Finally, the reviewer asks the question about the relationship between cell deformation and channel position. The reviewer is completely right, that the deformation in Figure 1 of the main manuscript reveals the same value at the inlet and inside the channel despite different hydrodynamic stress values. This observation originates from the fact, that the circularity as a one-dimensional parameter does not capture the full shape information of the cell. In practice, this can lead to situations where an ellipsoid-like shape at the inlet has the same deformation as a bullet-like shape inside the channel.

A full deformation characterization is e.g. included in the strain tensor or the Fourier components (Figure 2) but for simplicity we map this rather complex multi-dimensional parameter onto a single circularity value, which is easily accessible. This allows for a direct comparison of single cells based on a single number. For extracting an elastic Young's modulus we apply the results from ref. ¹ where steady-state traces yielding a bullet-like shape can be described analytically. As the complex hydrodynamics outside the channel is not covered by this model and in fact not needed for extracting rheological properties this approach is not a limitation and simplifies data evaluation. The validity of this framework is confirmed by calibration experiments with standardized microgel beads (please compare answer to next question).

Reviewer:

Finally but importantly, the study lacks control and validation experiments, e.g. measurements at different flow speeds to test for linearity, or measurements with elastic beads or oil droplets.

Author response:

We thank the reviewer for this very important suggestion and would like to point to Question 5 of reviewer 1. Following the idea of using elastic beads we have performed dynamic RT-DC measurement with poly-acrylamide microgel beads of a fixed elastic modulus following ref. ⁵. In this recent publication the authors demonstrate the generation of monodisperse beads with adjustable elasticity. For calibration beads with a polymer content of 7.9% (w/v) and a corresponding elastic modulus of 1.5 ± 0.5 kPa were obtained from Salvatore Girardo the main author of the publication.

Having a mean diameter of 15.2 ± 0.8 μm dRT-DC on these beads was performed in $20 \mu\text{m} \times 20 \mu\text{m}$ and $30 \mu\text{m} \times 30 \mu\text{m}$ channels at a flow rate of 16 nl/s and 128 nl/s, respectively. Following the protocol provided in the publication a deformation of approximately $d=0.02$ was adjusted before we applied our analytical model to extract the elastic modulus. For both, $20 \mu\text{m} \times 20 \mu\text{m}$ and $30 \mu\text{m} \times 30 \mu\text{m}$ channels, we obtain an elastic modulus of 2.20 ± 0.03 kPa and 2.00 ± 0.04 kPa, respectively (Supplementary Fig. 11).

These values are slightly higher than the reported results of 1.5 ± 0.5 kPa but within or close to the standard error of the mean. Two possible reason can be attributed to that. First of all the bead production reveals a slide batch-batch variation as shown in Table S4 of ref ⁵. Second, dRT-DC specifically incorporates the initial shape of the beads leading a reduced deformation magnitude compared to RT-DC and by that to a slightly elevated and more correct apparent elastic Young's modulus.

The authors from Girardo *et al.* have also been measuring the relaxation time of the ensemble average over many beads inside the channel and obtained a value of 0.12 ± 0.02 ms⁵. Remarkably, this is close to the value of 0.36 ± 0.01 ms (20 μ m x 20 μ m channel) and 0.12 ± 0.01 ms (30 μ m x 30 μ m channel) for single bead rheology in dRT-DC. In summary, these results confirm the possibility of calibrating dRT-DC and its utilization to extract rheological parameters from single cell measurements.

Based on this calibration we realized that under some conditions our standard analysis software (Shape-Out) extracts the wrong channel diameter from the measurement configuration files. This resulted in an overestimation of the apparent Young's modulus and subsequently the apparent viscosity. This made us re-analysing the data shown in Figure 3 and Figure 4 of the main manuscript. While deformation values and characteristic times are not affected, applying a channel diameter which is off by a factor of up to 1.5 (20 μ m x 20 μ m instead of 30 μ m x 30 μ m) resulted in an error in the magnitude of the elastic Young's modulus and viscosity. We would like to emphasize that this conversion error had only an impact on the absolute amplitude of the rheological parameters but not on the observed statistical trends. Since statistical analysis is based on linear mixed models comparing replicates of one group of samples to another group these relative differences are not affected.

We thank the reviewer for this great suggestion of using beads for calibration, we corrected our error and have been addressing the calibration of dRT-DC by adding the following paragraph on page 12 of the Methods section:

“Calibration of dynamic RT-DC

Calibration of dRT-DC has been carried out using standardized elastic poly-acrylamide microgel beads from ref.⁴⁸. The rheology of beads with a mean diameter of 15.2 ± 0.8 μ m and an elastic modulus of 1.5 ± 0.5 kPa has been characterized in 20 μ m x 20 μ m and 30 μ m x 30 μ m channels at a flow rate of

16 nl s⁻¹ and 128 nl s⁻¹, respectively. Using the peak channel deformation \hat{d}_{channel} and applying our analytical model we obtain an apparent elastic Young's modulus of 2.20 ± 0.03 kPa and 2.00 ± 0.04 kPa (Supplementary Fig. 11). The slight deviation from the calibration standard is still within or close to the standard error of the mean and could be attributed to batch-to-batch variation as discussed in ref⁴⁸. Interestingly, the authors from Girardo *et al.* also performed measurements on the relaxation time using ensemble averages over hundreds of beads. They find a value of 0.12 ± 0.02 ms, which is remarkably close to 0.36 ± 0.01 ms (20 μ m x 20 μ m channel) and 0.12 ± 0.01 ms (30 μ m x 30 μ m channel) for single beads.”

Reviewer #3 (Remarks to the Author):

The authors report the development of time-dependent measurements using the real-time deformability cytometry platform. By using Fourier analysis of the deformed shape of a cell as it transits in a channel with a viscous fluid they were able to extract two separate time-dependent deformation metrics, which they tie to physical parameters of cells using a fluid stress model. The authors are able to measure the viscoelastic properties of red blood cells, granulocytes, and lymphocytes in a single experiment which allows improved comparisons. Interestingly, at the population level differences in the mean elasticity and viscosity of B-lymphocytes and CD4+ T-lymphocytes was observed. Overall, this is a solid work, however, the final claim that population level differences in these lymphocytes could lead to accurate sorting, which requires classification at the single-cell level is not supported by the data.

Major comments:

1) In the final paragraph of the discussion the authors write: “In this context, we provide evidence that dRT-DC is capable to discriminate between isolated leukocytes of different lineages including B- and T-lymphocytes with statistical significance.” The authors then go on to discuss how this may be important to sorting. The authors should be more careful with these claims. Accurate sorting requires classification at the single-cell level, however, the authors only report that at the population level the means of the two populations are slightly different. However, no histograms of the populations are provided or some analysis of classification accuracy (ROC, AUC of the ROC, classification accuracy etc.). In order to make claims of distinguishing between single-cells between these populations this additional data should be provided. Alternatively the authors should only claim what is supported by the data, that the means of the population are statistically different, which does not apply to classification accuracy.

Author response:

We very much appreciate the positive recommendations of the reviewer. Following the suggestion, we first analysed the histograms of parameters extracted from dRT-DC and found some overlap as expected. The results are shown in Figure R6.

Figure R6: Parameter distribution for primary blood cells. The histograms show the distributions of (a) area, (b) elastic Young's modulus and (c) viscosity for erythrocytes, granulocytes and peripheral blood cells (data obtained from Fig. 4).

Taken advantage of this multidimensional dataset we applied a multiparameter logistic model to estimate the classification accuracy of our method. Starting from a set of parameters (cell size A , elastic Young's modulus E and viscosity η) we tested all possible combinations and selected the model with the lowest Akeike information

criterion (AIC) from the datasets. Preference is given to the model with a lower parameter number if the difference in AIC is less than two. This analysis was performed for the classification of erythrocytes, granulocytes and peripheral blood mononuclear cells (PBMCs). For a comparison between granulocytes and PBMCs having the largest overlap in the histogram (Fig. R6) we find a combination of area and viscosity yielding the model with the lowest AIC and the lower number of parameters (Table R2)

Parameter	AIC	Cutoff	AUC
(1) A	318.77	0.30	0.87
(2) E	466.54	0.32	0.62
(3) η	452.98	0.32	0.71
(4) $A + E$	314.38	0.30	0.88
(5) $A + \eta$	310.98	0.32	0.88
(6) $E + \eta$	447.37	0.30	0.73
(7) $A + E + \eta$	310.26	0.30	0.88

Table R2: Parameter sets for logistic regression model for granulocytes and peripheral blood mononuclear cells. Logistic regression models have been tested for all combinations of cell size A , the apparent elastic Young's modulus E and the apparent viscosity η .

For this model we derived the receiver operating characteristics and calculated the sensitivity and specificity. The results highlight that PBMCs can be identified with a sensitivity of exceeding 80% (Area under the curve 0.8) in a mixed leukocyte population (Figure R7a).

Figure R7: Logistic model for primary blood cells. a) Sensitivity and specificity for peripheral blood mononucleated cells (PBMCs) in a mixed population of granulocytes and PBMCs and b) sensitivity and specificity for CD4+ T-cells in a mixed lymphocyte population. Model selection using a multi-parameter approach was based on the Akeike information criterion (AIC) analyzing cell size, elastic Young's modulus and viscosity for (a) and cell size, peak deformation at the inlet and inlet relaxation time for (b). For (a) a model incorporating cell size and viscosity yields the lowest AIC (area under the curve AUC=0.88 and probability cutoff 0.32), while the minimum in AIC for (b) is found for a two-parametric model of the peak deformation at the inlet and the inlet relaxation time (AUC=0.62 and probability cutoff 0.80) from the receiver operating characteristic.

For classification of lymphocytes we realized that measures from the inlet trace d_{inlet} yield higher significances than channel measures d_{channel} and derived parameters, elastic Young's modulus and viscosity (Fig. 4 and Supplementary Fig. 6). Interestingly, B- and CD4+ T- cells are the only cells in this study which can statistically better be discriminated using inlet parameters. Therefore, our logistic regression models focus on cell size A , peak inlet deformation \hat{d}_{inlet} and inlet relaxation time τ_{inlet} and find a combination of \hat{d}_{inlet} and τ_{inlet} yielding the lowest AIC (Table R3). Here, CD4+ T cells can be identified with a sensitivity of 76% for a cutoff probability of 0.8 (Area under the curve 0.62) in a mixed lymphocyte population (Figure R7b). The specificity is 40%, which is sufficient for cell enrichment with high sensitivity.

Parameter	AIC	Cutoff	AUC
(1) A	3737.27	0.80	0.58
(2) \hat{d}_{inlet}	3650.66	0.80	0.62
(3) τ_{inlet}	3745.98	0.81	0.49
(4) $A + \hat{d}_{\text{inlet}}$	3649.43	0.80	0.62
(5) $A + \tau_{\text{inlet}}$	3736.54	0.80	0.57
(6) $\hat{d}_{\text{inlet}} + \tau_{\text{inlet}}$	3644.10	0.80	0.62
(7) $A + \hat{d}_{\text{inlet}} + \tau_{\text{inlet}}$	3643.52	0.80	0.62

Table R3: Parameter sets for logistic regression model for B-cells and CD4+ T-cells.

Logistic regression models have been tested for all combinations of cell size A , the peak inlet deformation \hat{d}_{inlet} and the inlet relaxation time τ_{inlet} .

Using the results of the logistic model analysis we have removed the elastic Young's modulus and the viscosity of the B- and CD4+ T- cells from the insets in Figure 4 and replaced them by the inlet and channel peak deformation. The results of this classification are now also summarized in the results section of the main manuscript on page 7:

“Finally, we use a logistic model to assess the capability of dRT-DC to discriminate leukocytes on a single cell level. Following a multivariate approach incorporating cell size A , apparent elastic Young's modulus E , and apparent viscosity η the Akaike information criterion (AIC) is used for model selection (see Methods). Comparing granulocytes and peripheral blood mononucleated cells we find a two-parametric model of A and η having the lowest AIC, which identifies PBMCs with a sensitivity exceeding 80% and an area under the curve AUC=0.88 (Supplementary Fig. 7a and Supplementary Table 2). For classification of CD4+ T-cells we have been analyzing logistic models of cell size A , the peak deformation \hat{d}_{inlet} and the inlet relaxation time τ_{inlet} . Here, a two-parametric model of \hat{d}_{inlet} and τ_{inlet} reveals the lowest AIC with a sensitivity of 76% and an AUC=0.62 (Supplementary Fig. 7b and Supplementary Table 3).”

and the following part in the methods section on page 13. In addition, Table R2 and R3 have been included in the Supplementary Material as Supplementary Table 2 and 3 and Figure R7 has been included in the Supplementary Material as Supplementary Fig. 7:

“The predictive potential of dRT-DC has been estimated based on the parameters: cell size A , apparent elastic Young's modulus E , apparent viscosity η , the peak inlet deformation \hat{d}_{inlet} and inlet relaxation time τ_{inlet} using logistic regression models. By considering all possible parameter combinations of A , E and η for the classification of granulocytes and PBMCs and all combinations

of A , \hat{d}_{inlet} and τ_{inlet} for the classification of B- and CD4+ T-cells model selection is based on Akaike's information criterion (AIC) ⁵⁰. For models describing the discrimination between (a) granulocytes and PBMCs as well as between (b) B- and CD4+ T-cells AICs are reported in the Supplementary Material (Supplementary Table 2 and 3). The model with the lowest AIC is considered the best model while preference is given to the model with the lower number of parameters if the differences in AIC is less than two. Receiver operating characteristics (ROC) including the area under the ROC curve (AUC) are derived for each model and optimal cutoff values for sensitivity and specificity have been calculated for the intersection in the ROC curve where sensitivity equals specificity ⁵¹. Data is always presented for the model yielding the lowest AIC."

Following these results and the suggestion of the reviewer we have also modified the discussion about the discrimination between B- and CD4+ T-cells and changed the sentence:

"This is of essential importance for future sorting applications based on intrinsic material properties which require real-time operation at high-throughput."

to:

"This is of essential importance for cell enrichment based on intrinsic material properties which require real-time operation at high-throughput."

Reviewer:

2) In a related matter, in Fig. 4a-d standard error of the mean is used. How many cells are used in these calculations? How much do the distributions from different cell types overlap? As discussed, it can be misleading for applications in classification of individual cells to use the SEM of the population, as population distributions can overlap significantly but still have a different mean and tight SEM if n is large enough.

Author response:

We thank the reviewer for pointing towards the question of the relation between cell number and SEM in our measurements. We would like to refer to the answer of the previous question of the reviewer and Figure R7. Here, we show for erythrocytes, granulocytes and peripheral blood mononuclear cells the overlap for parameters from dRT-DC measurements. While granulocytes and PBMCs reveal the strongest overlap a multiparameter logistic model allows to accurately classify PBMCs with a sensitivity exceeding 80% on a single cell level. For CD4+ T cells and B cells the situation is similar as described above.

We have included the information on cell classification in the main manuscript as well in the Methods section as stated in the answer to previous question. In addition the sample number is now stated in the caption of Figure 4 as follows:

"Statistical analysis in (c) and (d) includes data from $n=381$ erythrocytes, $n=243$ granulocytes, $n=130$ PBMCs, $n=737$ B- cells and $n=3,079$ CD4+ T- cells."

Reviewer:

3) It also, looks like based on the methods section that only CD4+ T-lymphocytes were measured (not CD8+ T-lymphocytes). This should be clarified.

Author response:

We thank the reviewer for this very helpful suggestion. It is correct that only CD4+ T-lymphocytes have been used for our measurements. We have corrected that in the Introduction (page 2), the caption of Figure 4, on top of page 7 and in the discussion on page 8.

Reviewer:

4) An important reference for measuring viscoelastic properties in DC geometries is missing: Kumar et al. Biophysical Journal 111, 2039–2050, November 1, 2016. Many similar experiments are performed (e.g. cytochalasin D treatment), and comparisons should be made with this work.

Author response:

We thank the reviewer for pointing towards this essential publication. We have now included this reference into our discussion and added the following sentence on page 8:

“Depolymerization of filamentous actin by CytoD leads to a reduction in the apparent elastic Young’s modulus as reported earlier for HL60 and 3T3 cells^{10,11}. While these previous studies successfully captured changes in elasticity by applying power-law rheology no significant differences in the fluidity were observed. Here, using dRT-DC and applying a linear viscoelastic model also allows to link actin depolymerization to a reduction in the apparent viscosity.”

Reviewer:

5) How is stress defined as a function of distance in the channel along a certain streamline. This is not very accurate as the stress a cell experiences will depend on location in the channel and dimensions of the cell, especially since the cell approaches the dimensions of the channel. When the cell is displacing the fluid as it flows through the channel the stress on the cell surface may be much higher than shown. It seems simulations are only performed for a single cell diameter – this should be clarified?

Author response:

We gratefully acknowledge the question from the reviewer highlighting a relevant fact. Following Mietke A. *et al.* the hydrodynamic stress on the cell surface is defined by the normal and shear component¹. The hydrodynamic shear stress is defined as the product of shear rate and the shear-rate dependent dynamic viscosity of the medium. While shear-rates are being calculated for each position inside the channel and for all relevant flow rates and cell sizes (see Table R1) using finite element method simulations the dynamic viscosity is measured for shear rates up to 3,000 1/s

using a rheometer (MCR502, Anton Paar). For shear rates exceeding the experimental range of our rheometer we fit a power law to the respective data for extrapolation.

This procedure has been described in the Methods section (Hydrodynamic simulations) and has been highlighted in the Supplementary Material (Supplementary Fig. 8) but was not well explained in the context of our analytical model. We have now changed the sequence in the Methods section and first introduce the hydrodynamic simulations before describing how to calculate the apparent elastic Young's modulus and apparent viscosity.

In addition, Table R1 is now included into the Supplementary Material as Supplementary Table 1 and referenced in page 11 (Methods section) of the main manuscript as follows:

"The simulations are carried out on physics-controlled meshes of an extra fine element size defined by the COMSOL backend. Parametric and material sweeps are performed for numerous conditions with respect to sample and sheath flow, channel geometry and object sizes (Supplementary Material Table 1). The layout of the microfluidic chips has been reconstructed, and one quadrant simulated, thus, taking advantage of the two symmetry axes through the center line of the design. Walls have been simulated using the no-slip boundary condition, and laminar flows entering the inlets of sample and sheath were assigned specific rates under constant pressure conditions. At the outlet, backflow is suppressed, reflecting the experimental conditions."

The question about the impact of cell sizes approaching the channel diameter is very interesting. Here, backflow in the corner of the squared cross-section will be relevant for cell deformation. Since this regime is not covered by our model we always choose a channel diameter exceeding the cell size by a factor of two. Therefore, we are outside this regime.

The second part of the questions deals with the displacement of the cell perpendicular to the direction of flow and the influence of cellular shape on the flow field. We have looked at the relationship between rheological parameters and lateral channel position by analysing the lateral distribution of cells inside the channel. For the data shown in Figure 3 this leads to a normal distribution of cells perpendicular to the channel axis as shown in the Q-Q plots of Figure S10 for a) wildtype HL60 cells, b) HL60 cells after treatment with 0.25% (v/v) DMSO and c) HL60 cells after treatment with 1 μ M Cyto D (top row). Analysing the standard deviation this distribution (with respect to the channel centre axis) we find $\sigma=0.68 \mu\text{m}$ (wildtype), $\sigma=0.57 \mu\text{m}$ (DMSO control) and $\sigma=0.95 \mu\text{m}$ (CytoD) verifying the efficiency of the hydrodynamic focussing.

In addition, we analysed the apparent elastic Young's modulus and the apparent viscosity as a function of the channel position. Here the standard deviation of the elastic Young's modulus is found to $\sigma=0.30 \text{ kPa}$ (wildtype and DMSO control), $\sigma=0.26 \text{ kPa}$ (CytoD) and for the viscosity $\sigma=2.80 \text{ Pa}\cdot\text{s}$ (wildtype), $\sigma=2.73 \text{ Pa}\cdot\text{s}$ (DMSO control) and $\sigma=2.99 \text{ Pa}\cdot\text{s}$ (CytoD). Performing a signed-rank test we find no correlation between the channel position and the apparent elastic Young's modulus as well as the apparent viscosity (correlation coefficient $\rho < 0.09$). This confirms that there is no systematic offset for small displacements from the channel centre axis.

For addressing the question of position dependency we have added Supplementary Figure S10 in the Supplementary Material and included the following paragraph in the Methods section (Analytical model) on page 12:

“Application of this simple model to calculate the Young’s modulus requires an alignment of the cells in the center of the channel³¹. We verified this condition by analyzing the apparent elastic Young’s modulus and the apparent viscosity as a function of the cell displacement from the channel center and find a standard deviation $\sigma < 1 \mu\text{m}$ for all conditions. Using a Spearman rank correlation test we find correlation coefficients of $\rho < 0.09$ for all conditions which confirms that there is no correlation between the rheological parameters and lateral cell position (Supplementary Fig. 10).”

Reviewer:

6) In a related matter, the different velocities in Fig.4 suggest different paths / different stress distributions. How is this taken into account with calculating Young’s modulus and subsequent viscosity parameters which depend on Young’s modulus?

Author response:

We thank the reviewer for this question. It is correct, that the velocities of the cells shown in Fig. 4a are different which implies a different stress distribution on each cell. Please note, that we have chosen smaller viscosities for erythrocytes compared to peripheral blood cells as we wanted to make sure that the erythrocytes do not deform too much.

For extracting the apparent elastic modulus we have been calculating the mean hydrodynamic stress on the surface of a cell using a finite element method (COMSOL 5.3a). This stress distribution is extracted from a full hydrodynamic representation of our microfluidic system under steady-state conditions incorporating the corresponding cell sizes, channel geometries, flow rates and the shear-thinning behaviour of our measurement buffer (Table R1). Performing the above calculations for a sphere and a bullet-like object, we could also show that the small deformations observed in dRT-DC do not impact on the hydrodynamic stress distribution (Figure R4 and Figure R5).

In practice, for each condition shown in the manuscript we map the mean cell size, the channel diameter, the flow rate and the rheological properties of our measurement buffer onto a hydrodynamic stress on the cell surface. An analytical model published earlier allows us to couple this hydrodynamic stress to linear elasticity theory representing the material properties of a cell, to predict cell deformation and to obtain the apparent elastic Young’s modulus¹.

From the apparent elastic Young’s modulus the apparent viscosity is calculated following a simple Kelvin-Voigt model. This approach requires a characteristic time τ_{channel} which we obtain from the channel deformation trace.

Finally, the reviewer asks about the impact of lateral cell position (inside the channel) on the apparent elastic Young’s modulus and apparent viscosity. Following up on the previous question of the reviewer, we have shown for Fig. 3 (HL-60 cells wildtype, 0.25% (v/v) DMSO, 1 μM CytoD) that a signed-rank test does not highlight any correlation ($\rho < 0.09$) between the channel position and the apparent elastic Young’s modulus as well as the apparent viscosity. This indicates that there is no systematic offset for small displacements from the channel centre axis (Supplementary Fig. 10).

These aspects were not very well explained in the original version of the manuscript. For clarification, we added the following paragraph to the Method section on page 12 (Analytical model):

“For calculation of cell rheological properties, we use the channel deformation trace d_{channel} only. Here, cells reach a steady-state deformation \hat{d}_{channel} , which is defined by the balance of normal and shear forces inside the constriction. In this steady-state we derive the cellular surface stress using FEM simulations as described above. The apparent elastic Young’s modulus E is then calculated using an analytical model published earlier³¹. Briefly, by coupling the hydrodynamic stress distribution in the channel to linear elasticity theory the deformation of a suspended spherical object can be predicted and mapped to the experimental conditions. Calculations are being performed for the mean size of each cell type presented in this study including flow-rate dependency and accompanying shear-thinning behavior of the surrounding medium.

While d_{channel} does not represent a simple strain function, e.g. $\varepsilon=\Delta l/l$, due to the parabolic hydrodynamic stress distribution, it is sufficient to represent the dynamic response of the cell to the step stress inside the channel and to derive a relaxation time from the fit of the channel deformation trace d_{channel} to Equation 5.

This relaxation time τ_{channel} can be considered the characteristic time for the creep-compliance response of the cell to a constant stress σ_{channel} (dashed line in Fig. 3a, bottom) inside the channel. Here, an apparent viscosity η can be calculated using a simple Kelvin-Voigt model with:

$$\tau_{\text{channel}} = \frac{\eta}{E}, \quad (7)$$

where E is the apparent elastic Young’s modulus.

Application of this simple model to calculate the Young’s modulus requires an alignment of the cells in the center of the channel³¹. We verified this condition by analyzing the apparent elastic Young’s modulus and the apparent viscosity as a function of the cell displacement from the channel center and find a standard deviation $\sigma < 1 \mu\text{m}$ for all conditions. Using a Spearman rank correlation test we find correlation coefficients of $\rho < 0.09$ for all conditions which confirms that there is no correlation between the rheological parameters and lateral cell position (Supplementary Fig. 10).”

Other comments:

Reviewer:

Figure 2c and Figure 2d are out of order. This might be intended to be read in a clockwise fashion, in addition to keeping the widths of a-d and b-c the same, but it is confusing.

Author response:

We thank the reviewer for this suggestion. We have now changed the sequence and replaced the Figure.

Reviewer:

Figure S3 should be broken into two different plots. Two y-axes are used, but the points don't really occupy the same x-axis. Colocation of the series on the same plot invites comparison between the two series, but with only two points in a series, they can be made to look as similar or dissimilar as desired.

Author response:

We thank the reviewer for this suggestion. We have now split the new Supplementary Fig. 5 (previous Supplementary Fig. 3) into subplot a) and b).

Reviewer:

Given how the overlapping traces are shown in Figure 2b, it is difficult to understand aspects of the single-cell behavior. For example, for a given axial position, are the single-cells normally distributed? Do the traces cross over each other? The traces seem to move together with a given sigma about the mean, but the data would be more interesting without much trace crossover.

Author response:

We acknowledge the reviewer for bringing up this very important point. The traces shown in Figure 2b yield the shape evolution of the Fourier components along the axial channel position. While crossings in Fourier space point towards different weights of the components and do not necessarily imply trace crossing in deformation space, we followed the suggestion of the reviewer and analysed the distribution at several channel positions. The graphs in Figure R8 shows HL60 cells at the inlet, the center and the outlet of the channel.

Figure R8: Cell distribution along channel. Histograms and Q-Q plots of HL60 cells at inlet, center and outlet of channel. Measurements represent $n=1,580$ cells and have been carried out in a $30\ \mu\text{m} \times 30\ \mu\text{m}$ channel at a flow rate of $8\ \text{nl/s}$.

We find that cells at the inlet and center slightly deviate from a normal distribution and are slightly skewed towards positive position values. This is expected as the hydrodynamic focussing in front of the inlet funnels the cells into the central channel where we observe a narrowing down of the distribution towards the outlet. Here, we observe a normal distribution.

Interestingly, the small second peak at positive position values (for this specific sample) is maintained at the inlet and the center position of the channel. This points towards the conclusion that cell traces do not cross but narrow down to a normal distribution at the channel outlet as expected for laminar flow. In fact, the Reynolds number in our system is smaller than 1 ($Re=0.1$ for typical experimental values of $\rho=1,000\ \text{kg/m}^3$, $v=0.1\ \text{m/s}$, $d=0.1 \cdot 10^{-6}\ \text{m}$, $\eta=10 \cdot 10^{-2}\ \text{Pa}\cdot\text{s}$) with no turbulences which could potentially induce trace crossover. The uniformity of the flow is also reflected in the small standard deviation of our data of which is below $1\ \mu\text{m}$ (Supplementary Fig. 10). We therefore conclude that while we cannot exclude trace crossover completely, its impact on the overall sample distribution is minor. As cells are measured serially there is also no interaction that might perturb the flow profile.

1. Mietke, A. *et al.* Extracting Cell Stiffness from Real-Time Deformability Cytometry: Theory and Experiment. *Biophys. J.* **109**, 2023–2036 (2015).
2. Desprat, N., Richert, A., Simeon, J. & Asnacios, A. Creep function of a single living cell. *Biophys. J.* **88**, 2224–2233 (2005).
3. Wu, P.-H. *et al.* A comparison of methods to assess cell mechanical properties. *Nat. Methods* **15**, 491–498 (2018).
4. Trepap, X. *et al.* Universal physical responses to stretch in the living cell. *Nature* **447**, 592–595 (2007).
5. Girardo, S. *et al.* Standardized microgel beads as elastic cell mechanical probes. *J. Mater. Chem. B* **6**, 6245–6261 (2018).
6. Mokbel, M. *et al.* Numerical Simulation of Real-Time Deformability Cytometry To Extract Cell Mechanical Properties. *ACS Biomater. Sci. Eng.* **3**, 2962–2973 (2017).
7. Kollmannsberger, P. & Fabry, B. Linear and Nonlinear Rheology of Living Cells. *Annu. Rev. Mater. Res.* **41**, 75–97 (2011).
8. Burnham, K. P. & Anderson, D. R. *Model Selection and Multimodel Inference.* (2002).
9. Greiner, M. Two-graph receiver operating characteristic (TG-ROC): a Microsoft-Excel template for the selection of cut-off values in diagnostic tests. *J. Immunol. Methods* **185**, 145–146 (1995).
10. Guillou, L. *et al.* Measuring Cell Viscoelastic Properties Using a Microfluidic Extensional Flow Device. *Biophys. J.* **111**, 2039–2050 (2016).
11. Nyberg, K. D. *et al.* Quantitative Deformability Cytometry: Rapid, Calibrated Measurements of Cell Mechanical Properties. *Biophys. J.* **113**, 1574–1584 (2017).

REVIEWERS' COMMENTS:

Reviewer #1 (Remarks to the Author):

I have read the revised version and think that the authors have done a good job clarifying their paper. I understand better what they have measured and calculated and it seems reasonable. I hate to complain but while they have modified the main text to better explain the shape analysis, I still can't find in the main text or supplementary material how they extract an estimate of E from the shape analysis. I do understand that they estimate a viscosity using η/E as a typical relation time in a Kelvin-Voigt model. I am not close enough to the field of cell analyses to comment critically on the authors' conclusions regarding discrimination based on cell type. The paper is well written and few remarks below are provided for further clarification.

1) p. 1: "had so far been hampered by LACK OF a fast and robust measurement technique"

2) p. 3: "relies on the assumption of steady-state deformations as well as a spherical cell shape" – the cell can't be deformed and spherical at the same time. Do the authors mean that the initial shape is assumed to be spherical?

3) p. 3: "derive time-dependent material properties like the viscosity of single cells" – This phrase makes no sense to me since the viscosity is not necessarily a time-dependent property.

4) p. 3: "an elastic Young's modulus E" – Young's modulus is an elastic modulus so the language is redundant. I think I prefer the language elastic modulus or effective elastic modulus since in most introductory discussions Young's modulus refers to a pure elongational deformation. Note "Young's" is misspelled at the top of p. 6.

5) p. 6: The author's estimate an apparent viscosity using a Kelvin-Voigt model. They may wish to write somewhere that the influence of the external phase viscosity has been neglected, which is likely fine for the samples they utilize.

6) p. 6: "Next, we perform single cell rheology experiments" – One feature that may be questionable here is that the bulk rheology of the same may have changed since different cell concentrations are being handled. I am not sure this is quantitatively significant but it is a distinct change in this next part of their measurements relative to the discussion on p. 4-5.

7) Methods: The two equations given for viscosities really should be rewritten to properly recognize units. As presented they are terrible can't have any generality. I assume that whatever was done with these equations is correct but as presented they are awful and suggest the authors do not know what they are talking about when it comes to equations and mechanics (which I do not believe is true since the authors' work strikes me as very good and original).

Reviewer #2 (Remarks to the Author):

The authors have addressed my questions and concerns.

Reviewer #1 (Remarks to the Author):

Reviewer:

I have read the revised version and think that the authors have done a good job clarifying their paper. I understand better what they have measured and calculated and it seems reasonable. I hate to complain but while they have modified the main text to better explain the shape analysis, I still can't find in the main text or supplementary material how they extract an estimate of E from the shape analysis. I do understand that they estimate a viscosity using η/E as a typical relation time in a Kelvin-Voigt model. I am not close enough to the field of cell analyses to comment critically on the authors' conclusions regarding discrimination based on cell type. The paper is well written and few remarks below are provided for further clarification.

Author response:

Thank you very much for the positive analysis of our revised version. We acknowledge the suggestion for a more detailed explanation of how the elastic modulus is extracted from the shape analysis. Currently, we briefly explain that in the Methods section on page 12 as follows:

“For calculation of cell rheological properties, we use the channel deformation trace d_{channel} only. Here, cells reach a steady-state deformation \hat{d}_{channel} , which is defined by the balance of normal and shear forces inside the constriction. In this steady-state we derive the cellular surface stress using FEM simulations as described above. The apparent elastic Young's modulus E is then calculated using an analytical model published earlier³¹. Briefly, by coupling the hydrodynamic stress distribution in the channel to linear elasticity theory the deformation of a suspended spherical object can be predicted and mapped to the experimental conditions. Calculations are being performed for the mean size of each cell type presented in this study including flow-rate dependency and accompanying shear-thinning behavior of the surrounding medium.”

For a better understanding we added the following sentence:

“For calculation of cell rheological properties, we use the channel deformation trace d_{channel} only. Here, cells reach a steady-state deformation \hat{d}_{channel} , which is defined by the balance of normal and shear forces inside the constriction. In this steady-state we derive the cellular surface stress using FEM simulations as described above. The apparent Young's modulus E is then calculated using an analytical model published earlier³⁰. Briefly, by coupling the hydrodynamic stress distribution in the channel to linear elasticity theory the deformation of a suspended spherical object can be predicted. Carrying out this analysis for a range of cell sizes and Young's modulus allows for generation of a look-up table, where experimental and predicted deformations can be compared. The stress distribution on the cell surface is derived from flow rate, channel diameter and viscosity for the mean size of each cell type presented in this study including flow-rate dependency and accompanying shear-thinning behavior of the surrounding medium.”

Reviewer:

1) p. 1: “had so far been hampered by LACK OF a fast and robust measurement technique”

Author response:

We thank the reviewer for this comment. We have changed the sentence accordingly.

Reviewer:

2) p. 3: “relies on the assumption of steady-state deformations as well as a spherical cell shape” – the cell can’t be deformed and spherical at the same time. Do the authors mean that the initial shape is assumed to be spherical?

Author response:

We thank the reviewer for pointing out this inconsistent description. The reviewer is correct, when stating that we refer to the initial cell shape. We have modified the sentence from:

“However, RT-DC is limited to a static region-of-interest (ROI) at the rear part of the microfluidic channel, relies on the assumption of steady-state deformations as well as a spherical cell shape, and cannot track the dynamics of shape changes, which would be a prerequisite to derive time-dependent material properties like the viscosity of single cells.”

to:

“However, RT-DC is limited to a static region-of-interest (ROI) at the rear part of the microfluidic channel, relies on the assumption of steady-state deformations as well as an **initially** spherical cell shape, and cannot track the dynamics of shape changes, which would be a prerequisite to derive time-dependent material properties like the viscosity of single cells.”

Reviewer:

3) p. 3: “derive time-dependent material properties like the viscosity of single cells” – This phrase makes no sense to me since the viscosity is not necessarily a time-dependent property.

Author response:

We acknowledge the reviewer for highlighting this important point. It is correct, that the viscosity is not necessarily time-dependent and it is also not what we wanted to formulate. For emphasizing that we can obtain elastic as well as viscous material parameters we changed the sentence from:

“However, RT-DC is limited to a static region-of-interest (ROI) at the rear part of the microfluidic channel, relies on the assumption of steady-state deformations as well as an initially spherical cell shape, and cannot track the dynamics of shape changes, which would be a prerequisite to derive time-dependent material properties like the viscosity of single cells.”

to:

“However, RT-DC is limited to a static region-of-interest (ROI) at the rear part of the microfluidic channel, relies on the assumption of steady-state deformations as well as an initially spherical cell shape, and cannot track the dynamics of shape changes. **This would be a prerequisite to derive elastic and viscous** material properties of single cells.”

Reviewer:

4) p. 3: “an elastic Young’s modulus E” – Young’s modulus is an elastic modulus so the language is redundant. I think I prefer the language elastic modulus or effective elastic modulus since in most introductory discussions Young’s modulus refers to a pure elongational deformation. Note “Young’s” is misspelled at the top of p. 6.

Author response:

The reviewer is right when stating that elastic Young’s modulus describes the same parameter twice and we thank the reviewer for highlighting that. We consistently now use the term ‘apparent Young’s modulus’. We would like to specify it in this way since elastic modulus could also include the term shear modulus for example. We also thank the reviewer for marking the spelling mistake which we have corrected.

Reviewer:

5) p. 6: The author’s estimate an apparent viscosity using a Kelvin-Voigt model. They may wish to write somewhere that the influence of the external phase viscosity has been neglected, which is likely fine for the samples they utilize.

Author response:

We thank the reviewer for this comment. It is correct that the external phase has been neglected since we assume a constant stress on the cell inside the channel. This assumption is justified as the deformation reconstructed from odd Fourier coefficients only is a response to the constant stress inside the channel. We have outlined that, e.g. in the following sentence on page 4:

“The response of the cell d_{channel} to the constant stress σ_{channel} inside the constriction is reconstructed from a_0 and the first five odd shape modes.”

Reviewer:

6) p. 6: “Next, we perform single cell rheology experiments” – One feature that may be questionable here is that the bulk rheology of the same may have changed since different cell concentrations are being handled. I am not sure this is quantitatively significant but it is a distinct change in this next part of their measurements relative to the discussion on p. 4-5.

Author response:

We thank the reviewer for pointing to the possibility of bulk rheology. The fact that the bulk rheology might change is correct. However, we are measuring single cells, which makes our assay independent from the initial concentration.

Reviewer:

7) Methods: The two equations given for viscosities really should be rewritten to properly recognize units. As presented they are terrible can’t have any generality. I assume that whatever was done with these equations is correct but as presented they are awful and suggest the authors do not know what they are talking about when it comes to equations and mechanics (which I do not believe is true since the authors’ work strikes me as very good and original).

Author response:

We thank the reviewer for highlighting this incorrect presentation and we apologize for the confusion. We have now corrected the equation from:

“The fluid is set to be incompressible and to the measured density of 1065 kg/m³ (DMA4500, Anton Paar). The shear-rate dependent viscosities $\eta_{1/2}(\dot{\gamma})$ of our measurement buffers have been characterized using a rheometer (MCR502, Anton Paar) and then modelled for PBS-/- with 0.6 % (w/v) methylcellulose by a simple power law to:

$$\eta_1(\dot{\gamma}) = 0.16 \cdot \dot{\gamma}^{(0.74-1)}$$

and for PBS-/- with 1 % (w/v) methylcellulose:

$$\eta_2(\dot{\gamma}) = 0.60 \cdot \dot{\gamma}^{(0.64-1)}$$

where $\dot{\gamma}$ denotes the hydrodynamic shear rate (Supplementary Fig. 8).”

to:

“The fluid is set to be incompressible and to the measured density of 1,065 kg m⁻³ (DMA4500, Anton Paar). The shear-rate dependent viscosities $\eta_{1/2}(\dot{\gamma})$ of our measurement buffers have been characterized using a rheometer (MCR502, Anton Paar) and then modelled for PBS-/- with 0.6 % (w/v) methylcellulose by a simple power law to:

$$\eta_1(\dot{\gamma}) = K_1 \cdot \left(\frac{\dot{\gamma}}{\dot{\gamma}_0}\right)^{(n_1-1)}, \#(5)$$

where K_1 is a consistency coefficient with $K_1 = 0.16$ Pa s and n_1 is the flow behavior index with $n_1 = 0.74$.

For PBS-/- with 1 % (w/v) methylcellulose we find:

$$\eta_2(\dot{\gamma}) = K_2 \cdot \left(\frac{\dot{\gamma}}{\dot{\gamma}_0}\right)^{(n_2-1)}, \#(6)$$

with $K_2 = 0.60$ Pa s and $n_2 = 0.64$. In both equations $\dot{\gamma}$ denotes the hydrodynamic shear rate and $\dot{\gamma}_0 = 1 \text{ s}^{-1}$ (Supplementary Figure 9).”

Reviewer #2 (Remarks to the Author):

Reviewer:

The authors have addressed my questions and concerns.

Author response:

We thank the reviewer for the positive feedback on the revised manuscript.